# An abundant quiescent stem cell population in *Drosophila* Malpighian tubules protects principal cells from kidney stones

Chenhui Wang, Allan C Spradling*

Howard Hughes Medical Institute Research Laboratories, Department of Embryology, Carnegie Institution for Science, Baltimore, United States

**Abstract** Adult *Drosophila* Malpighian tubules have low rates of cell turnover but are vulnerable to damage caused by stones, like their mammalian counterparts, kidneys. We show that *Drosophila* renal stem cells (RSCs) in the ureter and lower tubules comprise a unique, unipotent regenerative compartment. RSCs respond only to loss of nearby principal cells (PCs), cells critical for maintaining ionic balance. Large polyploid PCs are outnumbered by RSCs, which replace each lost cell with multiple PCs of lower ploidy. Notably, RSCs do not replenish principal cells or stellate cells in the upper tubules. RSCs generate daughters by asymmetric Notch signaling, yet RSCs remain quiescent (cell cycle-arrested) without damage. Nevertheless, the capacity for RSC-mediated repair extends the lifespan of flies carrying kidney stones. We propose that abundant, RSC-like stem cells exist in other tissues with low rates of turnover where they may have been mistaken for differentiated tissue cells.

**\*For correspondence:**
spradling@carnegiescience.edu

**Competing interests:** The authors declare that no competing interests exist.

## Introduction

The *Drosophila* digestive system provides an outstanding model in which to identify novel mechanisms of tissue maintenance by stem cells (reviewed in *Losick et al., 2011*; *Miguel-Aliaga et al., 2018*). Highly active midgut intestinal stem cells (ISCs) produce regionally distinctive enterocytes and enteroendocrine cells throughout the midgut via asymmetric Notch signaling (*Filshie et al., 1971*; *McNulty et al., 2001*; *Dubreuil, 2004*; *Ohlstein and Spradling, 2007*; *Veenstra et al., 2008*; *Mehta et al., 2009*; *Shanbhag and Tripathi, 2009*; *Buchon et al., 2013*; *Marianes and Spradling, 2013*). Under normal conditions, ISCs respond to the demands of diet (*Choi et al., 2011*; *O'Brien et al., 2011*; *Obniski et al., 2018*), differences in spatial location (*Buchon et al., 2013*; *Marianes and Spradling, 2013*), mechanical forces (*He et al., 2018*; *Li et al., 2018*), the microbiota (*Buchon et al., 2009a*; *Buchon et al., 2009b*) and age (*Biteau et al., 2008*; *Choi et al., 2008*), all of which can influence the rate of cell turnover. In addition, ISCs support regenerative pathways that incorporate a broader range of cellular behavior when the digestive system is damaged by dietary toxins (*Amcheslavsky et al., 2009*; *Chatterjee and Ip, 2009*), or pathogens (*Buchon et al., 2009a*; *Buchon et al., 2009b*). Other regions of the gut are maintained and repaired in different ways. The hindgut lacks specialized stem cells and following damage is maintained primarily by induced polyploidization of post-mitotic epithelial cells (*Fox and Spradling, 2009*; *Losick et al., 2013*; *Sawyer et al., 2017*; *Cohen et al., 2018*).

The adult *Drosophila* excretory organ, the Malpighian tubules (MTs), comprise the fastest known fluid-transporting epithelium (*Maddrell, 2009*). MTs represent an anatomically simple adjunct of the digestive system consisting of initial, transitional, main and lower segments (*Figure 1A*, *Figure 1—figure supplement 1*). The four Malpighian tubules branch from two common ureters that drain into

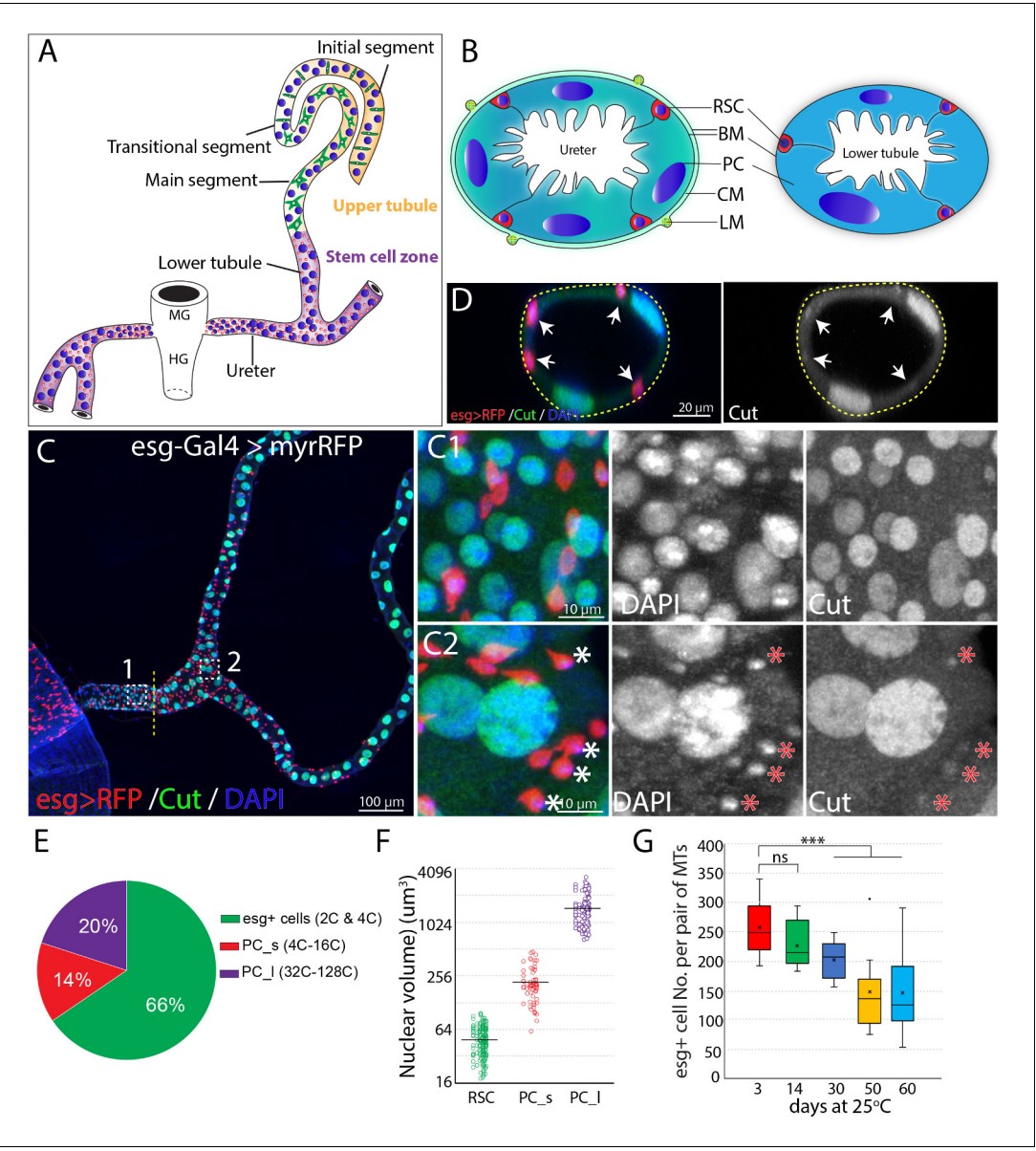

**Figure 1.** Adult *Drosophila* renal stem cells reside in the ureter and lower tubules. (A) Drawing showing an adult *Drosophila* Malpighian tubule with its connection at the midgut (MG)/hindgut (HG) junction. Stem cell zone (purple) comprises the ureter and lower tubules, upper tubules (yellow) consist of main segment, transitional segment and initial segment. Renal stem cells (red), principal cells (blue) and stellate cells (green) are indicated. (B) Drawing of cross section of the ureter (left) and lower tubule (right). RSC: renal stem cell; BM: basement membrane; PC: principal cell; CM: circular muscle; LM: longitudinal muscle. (C) Z-stacked immunofluorescence micrograph of the ureter and lower tubules from a 3 day old *esg-Gal4 >UAS* myrRFP female. The lower ureter containing smaller PCs joins with the upper ureter containing large PCs as indicted by a dotted line. Regions of both C1 and C2 are shown at high magnification on the right. Note relatively weak staining of Cut was seen in RSCs (denoted by asterisks). (D) A cross section view of lower tubule. RSCs are indicated by arrows. (E) Graph summarizing the relative numbers of the major cell types in the ureter and lower tubules. PC_s: small principal cells located at the lower ureter; PC_l: large principal cells located at the upper ureter and lower tubules. n = 10 3–5 days old animals. (F) Plot of nuclear volume of the indicated cells from the stem cell zone, showing differences that strongly correlate with ploidy. (G) Box plot showing the number of RSCs per pair of Malpighian tubules in 3,14,30,50 and 60 day-old animals. n = 8–29 animals.. *** denotes Student's t test p<0.001, ns denotes not significant with p>0.05. For (E), (F) and (G), see also *Figure 1—source data 1*.

The online version of this article includes the following source data and figure supplement(s) for figure 1:

**Source data 1.** Source data for *Figure 1E-G*.

*Figure 1 continued on next page*

*Figure 1 continued*

**Figure supplement 1.** Adult *Drosophila* Malpighian tubules are organized into different compartments.

the gut at the midgut/hindgut junction. These tubular organs are essentially cellular monolayers that function as kidneys (*Figure 1B and D*). It is commonly believed that the main segment of Malpighian tubules is responsible for fluid secretion, whereas the ureter and lower tubules are reabsorptive (*Dow et al., 1994*; *O'Donnell and Maddrell, 1995*). The main segment of Malpighian tubules contains only two differentiated cell types, principal cells (PC) and stellate cells, which are responsible for regulating ion balance and fluid secretion (reviewed in *Gautam et al., 2017*).

MTs contain a distinct population of renal stem cells (RSCs) that express *escargot (esg)* (*Singh et al., 2007*) and arise from the same pool of adult midgut progenitors that generate posterior midgut ISCs (*Takashima et al., 2013*; *Xu et al., 2018*). RSCs are located both in the lower ureter, which develops from adult midgut progenitors, and in the upper ureter and lower segment of the Malpighian tubule (MT), which derive from the surviving larval ureter and larval MT (*Takashima et al., 2013*). RSCs were proposed to divide actively to produce daughter renoblasts (RBs) which migrate and endoreduplicate to replenish the two main MT cell types- principal cells (PCs) and stellate cells (SCs) located throughout the MT (*Singh et al., 2007*). However, follow-up studies demonstrated that RSCs are not active and divide slowly. Notch signaling and EGFR/MAPK signaling pathways regulate the differentiation and proliferation of RSCs, respectively (*Li et al., 2014*; *Li et al., 2015*). However, it remains unclear how RSCs contribute to regeneration of MTs after damage. *Drosophila* Malpighian tubules are potential models for multiple aspects of metazoan kidney function, including the role of stem cells in maintaining adult nephric function.

Here we used novel methods to discover that RSCs comprise an abundant population of quiescent (cell cycle-arrested), unipotent stem cells that act only locally, on nearby principal cells. They respond to tissue injury located nearby, when they exit quiescence by upregulating the JNK, EGFR/MAPK, Yki and JAK/STAT pathways. Injury also activates asymmetric Notch signaling and upregulates Cut expression, to ensure that RSC daughters differentiate into replacement PCs. Replacement PCs produced by RSCs average about 8-fold lower ploidy than starting PCs, but are produced in a correspondingly greater number to maintain tissue DNA content. RSCs aid survival when Malpighian tubules are damaged by kidney stones, highlighting the importance of RSC-mediated regeneration.

## Results

### Adult Malpighian tubules contain abundant RSCs in the lower segment and ureter that decline in number during adulthood

Initially, we characterized the cell population in the ureter and lower tubules to quantitate the relative numbers of RSCs, and PCs. Interestingly, two populations of principal cells are easily distinguishable based on size and ploidy: small PCs (4–16C) are found in the lower ureter, and large PCs (32–128C) reside in the upper ureter and Malpighian tubules (*Figure 1C and F*). Both small PCs and large PCs express the homeobox protein Cut at high levels (*Figure 1C1 and C2*). We confirmed that *esg*-positive diploid cells are present only in the ureter and lower tubule (which we term 'the stem cell zone (SCZ)') using a Gal4 enhancer trap of *esg* to drive the expression of UAS-RFP (*Figure 1C*). Intriguingly, the *esg*⁺ progenitor cells make up about 66% of cells in the SCZ, more than the sum of small and large PCs (*Figure 1C and E*). We also detected a low expression level of Cut in *esg*⁺ stem cells (*Figure 1C2 and D*), consistent with a recent study (*Xu et al., 2018*). We found that the *esg*⁺ cell population slowly declines in number during adulthood (*Figure 1G*).

### RSCs resemble ISCs in gene expression

To investigate the gene expression in the cells of the ureter and lower tubule, especially RSCs, we performed single cell RNA-seq (scRNA-seq) to profile the transcriptome of disassociated tubule cells from the isolated ureters and lower tubules of young wild type female flies. 710 cells were sequenced and retained after quality control. The median genes that were recovered per cell is 1611. We used Seurat (*Stuart et al., 2019*) to do cell cluster assignment and four major and one minor cell clusters were resolved as shown in the uniform manifold approximation and projection

(UMAP) plot (*McInnes et al., 1802*; *Figure 2A*). The four major cell groups correspond to RSCs (group 1) and three principal cell subtypes (group 2–4) based on expression of known markers (*Figure 2B and C*). The presence of a small population of Drip[+] cells (group 5) is most likely due to contamination of stellate cells during dissection since stellate cells in vivo are only found in the upper tubules (*Figure 1—figure supplement 1A and F*).

RSCs were assigned to group 1 cells based on the enriched expression of *esg*. Further analysis revealed that group 1 cells expressed many genes such as *Delta, Notch,* and *Sox21a* that are important for ISC regulation, and antibodies specific for Dl and Notch labeled RSCs (*Figure 2D, E and G*), consistent with the previous studies (*Li et al., 2014*). Although both Dl and Notch are present in *esg*[+] RSCs, Notch signaling activity was barely detectable in the adult Malpighian tubules, but readily detectable in midgut (*Figure 2F*), using a Notch signaling reporter construct expressing GFP under a Notch response element (NRE-GFP) (*Saj et al., 2010*).

By comparing transcription factor expression in group 1 RSCs with published data for ISCs to gain potential insight into cell identity (*Dutta et al., 2015*), we found that RSCs were strikingly similar to ISCs. Relatively highly expressed TFs in RSCs were *esg, Sox21a, E(spl)malpha-BFM, E(spl) mbeta-HLH, E(spl)m3-HLH, Sox100B, bun, zfh2, fkh, peb, Myc, Eip93F, Pdp1, Eip75B,* and *CrebA*. All of these genes are also expressed in ISCs. The chromatin decondensation factor 31 (*Df31*), which is associated with the establishment of open chromatin (*Schubert et al., 2012*), is highly enriched in RSCs (*Figure 2H*) and is also strongly expressed in ISCs (*Dutta et al., 2015*) (FlyGut-seq, http://fly-gutseq.buchonlab.com/).

## Subtypes of principal cells in the SCZ

The principal cells (PC1-PC3) were assigned to clusters 2–4 based on the high expression of known principal cell marker genes including *ct, Alp4, Uro* and *salt* (*Wallrath et al., 1990*; *Sözen et al., 1997*; *Wang et al., 2004*; *Stergiopoulos et al., 2009*). PCs also expressed *Ndae1, NaPi-T, Eglp2, Vha100-2, Irk2, Irk3* and *ZnT35C*, genes involved in ion transport. The gene *caudal* (*cad*), which encodes a homeobox domain transcription factor essential for normal hindgut formation (*Lh and Lengyel, 1998*), was also expressed in the PC1-PC3 cells (*Figure 2C and I*). As further validation, a Gal4 enhancer trap line derived from the *caudal* (*cad)* gene specifically labeled principal cells in the ureter and the lower tubules (*Figure 2I*). Further analysis of differentially expressed genes among clusters 2–4 revealed that they correspond to PC subtypes. The relatively lower expression of *Alp4* and *cad* in PC1 cells suggested they are the principal cells in the distal lower tubules and proximal main segments, whose principal cells highly express *Uro* and *salt* (*Wallrath et al., 1990*; *Stergiopoulos et al., 2009*). The differential expression of *Pvr* between PC2 and PC3 cells enabled us to tell them apart (*Figure 2C*). A Pvr-GFP reporter construct was highly expressed in the principal cells of the lower ureter (*Figure 1—figure supplement 1C*). Thus PC3 cells correspond to the small principal cells in the lower ureter, whereas PC2 cells correspond to the large principal cells in the upper ureter and proximal lower tubules.

## Renal stem cells replenish principal cells in the SCZ but do not replenish stellate or principal cells in the upper tubules

In animals, tissue resident stem cells produce differentiated cells to maintain tissues in normal and/or stress conditions. To determine whether RSCs can respond to principal cell loss in the SCZ, we genetically ablated PCs by conditionally driving the expression of the apoptosis-inducing genes *reaper* and *hid* (*Zhou et al., 1997*). Forced expression of *UAS-hid, UAS-rpr* for 7 days at 29℃ with *c507-Gal4, tub-Gal80*[ts] (referred to as *c507-Gal4*[ts]), which is a Gal4 enhancer trap line of Alp4 and specifically marks PCs in the SCZ (*Yang et al., 2000*; *Figure 1—figure supplement 1D*), led to nearly complete ablation of preexisting large PCs in the SCZ. The regenerating SCZ was infiltrated with supernumerary small cells (*Figure 3A and B*). However, after 21 days of recovery at 18℃, most small cells were gone and cells that were present appeared to be new PCs with high expression levels of Cut (*Figure 3C*, *Figure 3—figure supplement 1A–B*). Previous studies have shown that stellate cells express the zinc-finger transcription factor *teashirt* (*tsh*) (*Sözen et al., 1997*; *Denholm et al., 2003*). The replacement cells did not express the *tsh* as revealed by the lacZ enhancer trap line (*tsh*[04319]) (*Figure 3—figure supplement 1C–E*), indicating they do not adopt stellate cell fate. Notably, the replacement PCs have significantly smaller nuclei compared to preexisting

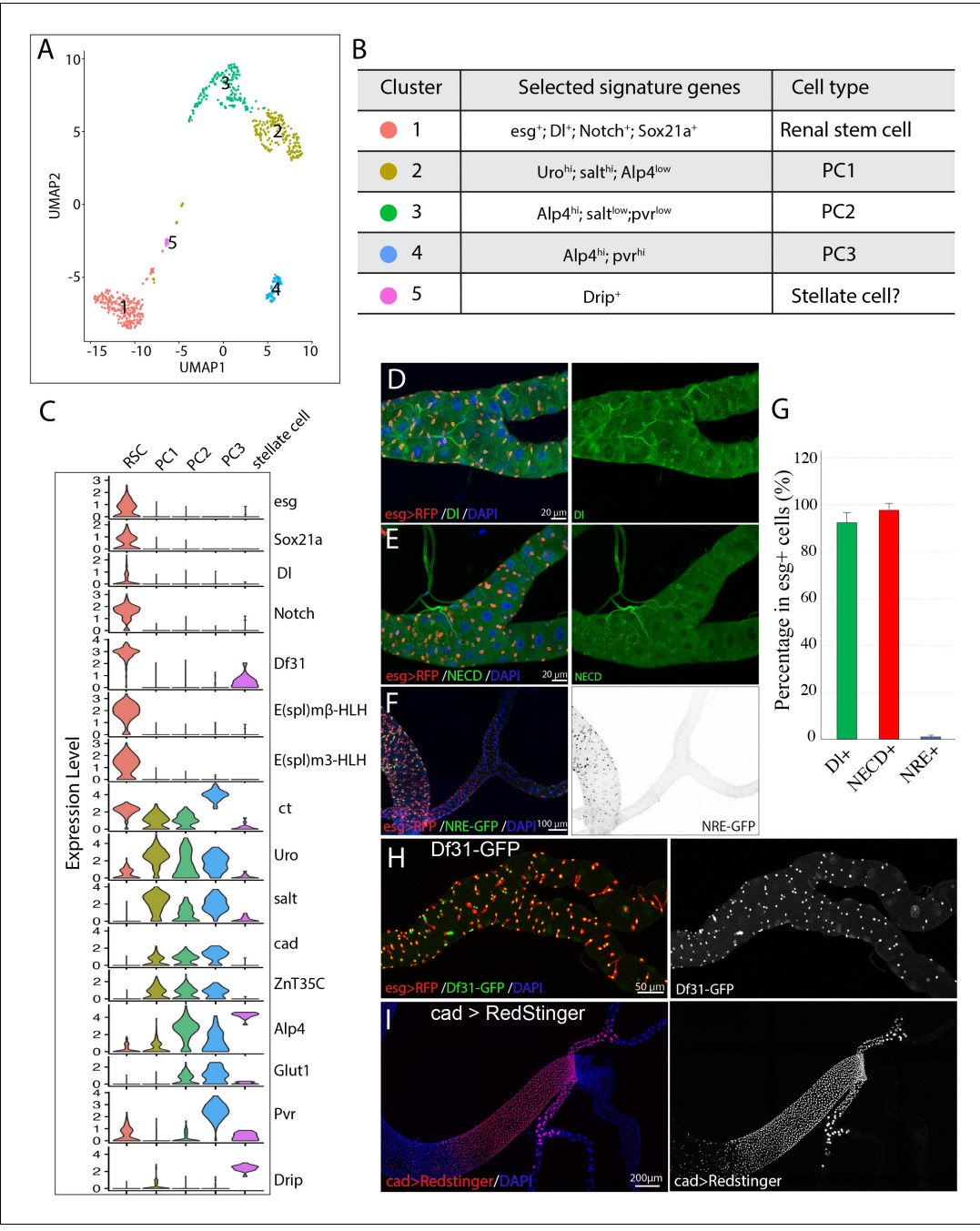

**Figure 2.** Single cell gene expression of the SCZ in adult Malpighian tubule. (**A**). Uniform Manifold Approximation and Projection (UMAP) plot of tubule cells in the SCZ of adult Malpighian tubules. (**B**) Cell types are identified based on differentially expressed marker genes. (**C**) Violin plots showing the expression of selected genes among the five cell types. (**D**). Immunostaining of Dl showing Dl is expressed in RSCs. (**E**) Immunostaining of the Notch extracellular domain (NECD) showing Notch is expressed in RSCs. (**F**) Immunofluorescence micrograph showing NRE-GFP is barely detectable in adult Malpighian tubules. (**G**) Quantification of percentage of RSCs that express Dl, Notch and NRE-GFP, respectively. Data are means ± SD, n = 10–11 animals. See *Figure 2—source data 1*. (**H**) Expression of Df31-GFP is highly enriched in RSCs. (**I**) Expression of *cad* visualized using *cad-Gal4* driven UAS-RedStinger expression.

The online version of this article includes the following source data for figure 2:

**Source data 1.** Source data for *Figure 2G*.

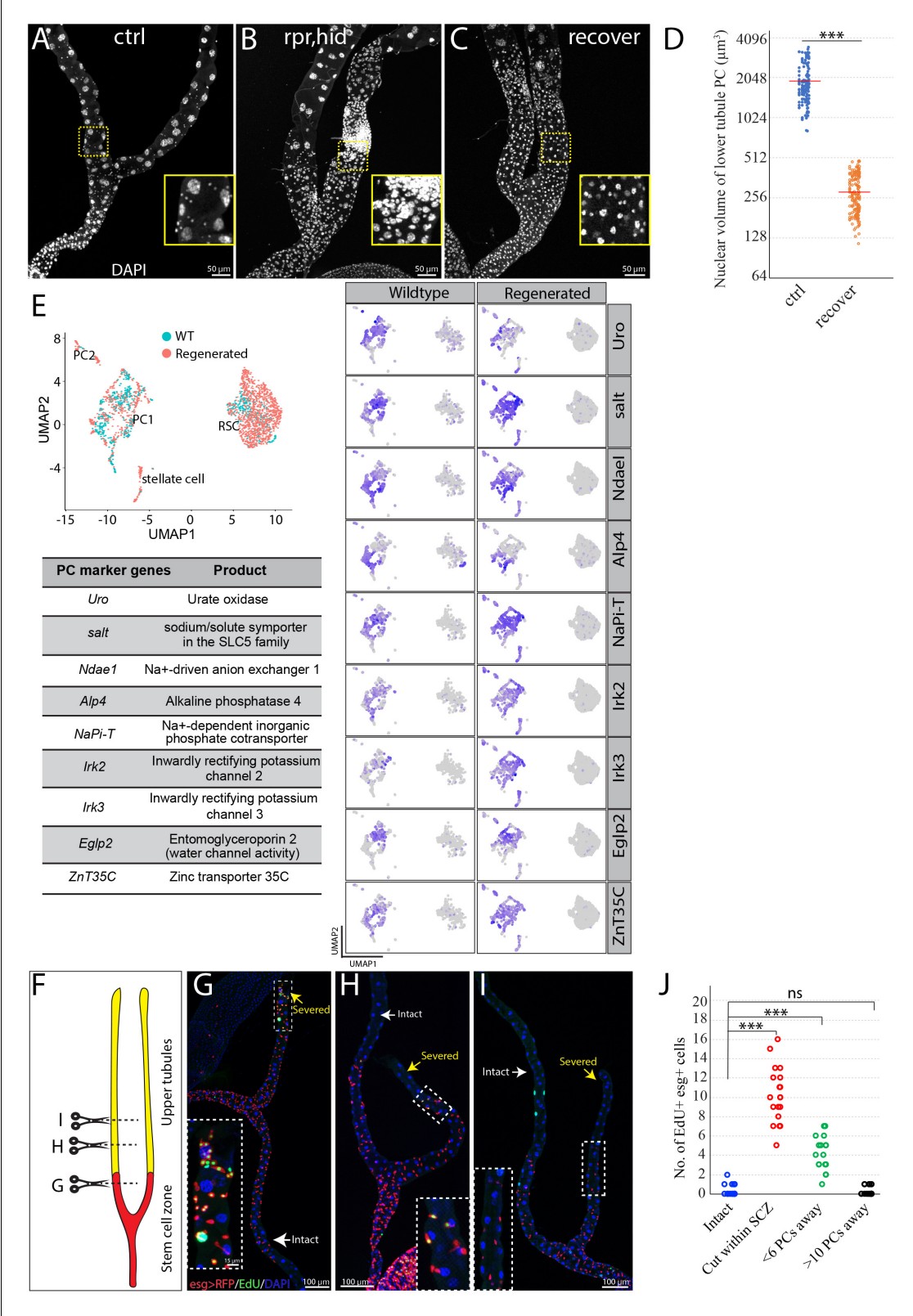

**Figure 3.** RSCs replenish principal cells in the SCZ upon damage. (**A**) Control (*c507-Gal4^ts*>RFP ). (**B**) Expression of *UAS-rpr, hid* with *c507-Gal4^ts* at 29°C for 7 days ablated preexisting PCs in the SCZ. (**C**) Recovery at 18°C for 21 days following PC-ablation. (**D**) Quantification of nuclear volume of preexisting PCs and replacement PCs in the SCZ from control animals and recovered animals, respectively. Bar denotes the average value. (**E**) Integrated analysis of scRNA-seq data from wild type and regenerated SCZ following PC-ablation. 710 cells from wild type and 1499 cells from

*Figure 3 continued on next page*

*Figure 3 continued*

regenerated tissues were retained and integrated. Same cell types in wild type and regenerated SCZ were clustered together. The presence of stellate cell cluster is most likely due to contamination during dissection (upper tubule cells were not completely removed). Examination of nine highly expressed genes physiologically important for principal cells further indicates that the smaller replacement principal cells resemble the large pre-existing principal cells based on gene expression. (F) Schematic drawing depicting the surgical procedures to remove part of Malpighian tubules (See Materials and methods). Surgical sites in (G–I) are shown by dashed lines. (G) Surgical injury in the lower tubule promoted EdU incorporation into *esg* >*RFP*⁺ RSCs near the surgical site. (H) Surgical injury in the upper tubule 5 PCs away from the SCZ slightly increased EdU incorporation in the RSCs at the most distal SCZ. (I) Surgical injury in the upper tubule 10 PCs away from the SCZ did not increase EdU incorporation in the RSCs. (J) Quantification of EdU⁺ *esg*⁺ cell number per lower tubule after surgical resection at different sites. *** denotes Student's t test p<0.001, ns denotes not significant with p>0.05. For (D) and (J), see also *Figure 3—source data 1*.

The online version of this article includes the following source data and figure supplement(s) for figure 3:

**Source data 1.** Source data for *Figure 3D and J*.
**Figure supplement 1.** Ablation of principal cells in the ureter and lower tubules leads to formation of replacement principal cells.
**Figure supplement 2.** No replenishment of either stellate cells or principal cells in the upper tubules after ablation.
**Figure supplement 3.** Damage activates multiple proliferation pathways.

PCs in the lower tubules (*Figure 3D*). In spite of distinct cell size, the replacement PCs resemble pre-existing PCs in their transcriptomes as revealed by integrated analysis of scRNA-seq data across the wild type and regenerated tissues (*Figure 3E*).

It was previously reported that RSCs are multipotent stem cells that can replenish both principal cells and stellate cells throughout the Malpighian tubules (*Singh et al., 2007*). Since the upper tubules are about 1.5 mm long and are completely devoid of RSCs, we questioned whether RSCs can respond to tubule epithelial cell loss in the distal tubules. To investigate, we genetically ablated stellate cells or principal cells in the upper tubules using Gal4/UAS/Gal80ᵗˢ system to conditionally drive the expression of the apoptosis-inducing genes *reaper* and *hid*, and then monitored regeneration at 18°C to minimize further expression.

We first examined whether stellate cells which are only localized in the upper tubules could be replaced. The Gal4 enhancer trap line *tsh^md621* (*tsh-Gal4*) driven expression of UAS-RFP specifically labels stellate cells in Malpighian tubules (*Figure 1—figure supplement 1F*; *Figure 3—figure supplement 2A*). Expression of *UAS-hid, UAS-rpr* with *tsh-Gal4, tub-Gal80ᵗˢ* for 7 days at 29°C resulted in a near-complete ablation of stellate cells (*Figure 3—figure supplement 2B, C and E*). However, we did not observe any replenishment of stellate cells even 30 days after shifting the animals back to 18°C for recovery (*Figure 3—figure supplement 2D–E*). This result indicates that stellate cells are not replaced in adult flies and that RSCs do not respond to stellate cell loss.

We next investigated whether RSCs can respond to the loss of principal cells in the upper tubules. The *Gal4* under the control of putative promoter of *Uro* (referred to as *Uro-Gal4*) is specifically expressed in the principal cells in the main segment (*Terhzaz et al., 2010*), as revealed by the expression *UAS-myrRFP* driven by *Uro-Gal4*, no expression was detected in stellate cells (*Figure 3—figure supplement 2F*). We ablated principal cells in the main segment by expressing *UAS-hid, UAS-rpr* with *Uro-Gal4, tub-Gal80ᵗˢ*. After expressing *rpr* and *hid* 7 days at 29°C, most principal cells in the main segment were ablated (data not shown). After shifting the animals back to 18°C for recovery, we observed supernumerary ectopic tubule cells at the distal end of the stem cell zone. However, the ectopic tubule cells never migrated into the upper tubule to replace lost principal cells in the upper tubules (*Figure 3—figure supplement 2G*). These results indicate that RSCs may sense and respond to damage located very close to the SCZ, however, newly generated RSC daughters do not migrate into the upper tubules. Collectively, these results demonstrated that RSCs can replenish the PCs in the SCZ but lack detectable capacity to replenish stellate cells and PCs in the upper tubules.

## Damaged PCs within the SCZ are replaced by smaller cells

One characteristic of the new PCs generated in response to PC loss was a reduction in size. For example, when PCs in the SCZ were ablated by expression of *rpr* and *hid* for 7 days followed by recovery at 18°C for 21 days, nearly all the giant polyploid PCs were lost (*Figure 3A–C*). New PCs were generated throughout the SCZ that were about eight-fold lower in nuclear volume (*Figure 3D*).

To determine how sensitive RSCs are to the loss of principal cells in the Malpighian tubule, we developed a surgical procedure that allows the distal ends of Malpighian tubules to be severed in a living adult animal after pulling it through a small incision in the abdominal cuticle (*Figure 3F*, see Materials and methods). The cut end appears to close off and the operated animals remain viable and live a normal lifespan. When the cut end was located within the SCZ, DNA replication as measured by 5- ethynyl-2'-deoxyuridine (EdU) incorporation increased in nearby *esg*-positive RSCs, but not throughout the entire SCZ (*Figure 3G*). Damage in the main segment could only activate RSCs if it was close to the SCZ junction. Damage about five PCs away increased EdU incorporation in the most distal RSCs of the SCZ (*Figure 3H*). However, damage about ten PCs away did not increase EdU-labeled RSCs (*Figure 3I–J*), indicating it failed to activate RSCs. This suggests that RSCs sense damage through short range signals extending less than ten cells away along the tubule.

We investigated repair at the cellular level by lineage marking RSCs using *esg-Gal4* and *tub-Gal80^{ts}* (referred to as *esg-Gal4 ^{ts}*, see Materials and methods). Cutting a tubule within the SCZ and activating lineage tracing by temperature shift revealed a substantial increase in the number of RSC daughters (i.e. marked *esg^-*) cells near the surgical site (*Figure 3—figure supplement 1F and F'*, 1H), indicating that local RSCs were induced to generate *esg^-* daughters by surgical injury in the SCZ. In contrast, truncating the tubule within the main segment about 10 PCs away from the SCZ did not elicit any production of lineage-marked *esg^-*cells in the SCZ up to 21 days post surgery (*Figure 3—figure supplement 1G and G'*,1H).

## Damage activates quiescent RSCs via upregulation of multiple short range signaling pathways

To investigate what damage-induced signals activate the RSCs, we examined damage pathways known to activate ISC proliferation. Extensive studies have shown that the EGFR/MAPK, Jak/Stat, Wnt, Hippo, PVF, JNK, Insulin, and BMP pathways are all involved in eliciting ISC division either under steady state conditions or during regeneration following cellular damage (*Guo et al., 2016*; *Jiang et al., 2016*; *Li and Jasper, 2016*; *Gervais and Bardin, 2017*). Reporters for four of these pathways, JNK, Jak/Stat, MAPK and Yki pathways were induced upon surgical injury (*Figure 3—figure supplement 3*). Interestingly, the JNK pathway was induced as early as 12–24 hr after injury in cells at the surgical site as revealed by the expression of *UAS-GFP driven by puc^{E69}-Gal4* (*Adachi-Yamada, 2002*), regardless of whether the surgical site was within the SCZ or in the distal upper tubules (*Figure 3—figure supplement 3A*). However, upregulation of Jak/Stat and EGFR pathway reporters took place in RSCs close to the surgical site, which was only within the SCZ or in the proximal upper tubules, as assayed by the expression of 10XStat92E-GFP (*Bach et al., 2007*) and dpERK (*Gabay et al., 1997*), respectively (*Figure 3—figure supplement 3B and C*). *Diap1-lacZ*, which is widely used as reporter of Yki activity (*Huang et al., 2005*), was also induced in cells near the surgical site within the SCZ (*Figure 3—figure supplement 3D*), suggesting the Yki signaling was activated. JNK pathway activation can induce the expression of *Drosophila* Jak/Stat pathway ligands (Upd1, Upd2 and Upd3), which in turn transduce Jak/Stat signaling in neighboring cells (*Jiang et al., 2009*). We propose that RSCs are activated by damage-induced JNK activation, which is then relayed by short range signals including Upds and EGFs. This mechanism may explain why only local damage is sufficient to activate RSCs.

## Notch signaling is essential to generate differentiated RSC progeny

We next investigated the genes responsible for controlling the differentiation of replacement principal cells from activated RSCs. As in the case of ISCs, Notch signaling was found to play a critical and central role. Notch signaling was barely detectable in RSCs under normal conditions using NRE-GFP as a reporter. However, Notch activity was induced in cells close to the surgical site after MT resection (*Figure 4A*). Expression of Cut, a transcription factor that distinguishes renal from midgut tissue (*Xu et al., 2018*), was also elevated in the RSCs near the surgical site (*Figure 4B*).

In the midgut, disruption of Notch blocks intestinal stem cell daughter differentiation and leads to the proliferation of ISC-like cells (*Ohlstein and Spradling, 2007*). In the SCZ, we found that RNAi-mediated knockdown of Notch caused an accumulation of RSC-like cells specifically near the surgical site (*Figure 4C and D*), suggesting that it plays an important role in injury-activated RSCs but not normal RSCs. To further validate the finding, heat-shock induced $N^{55e11}$ mutant clones marked by

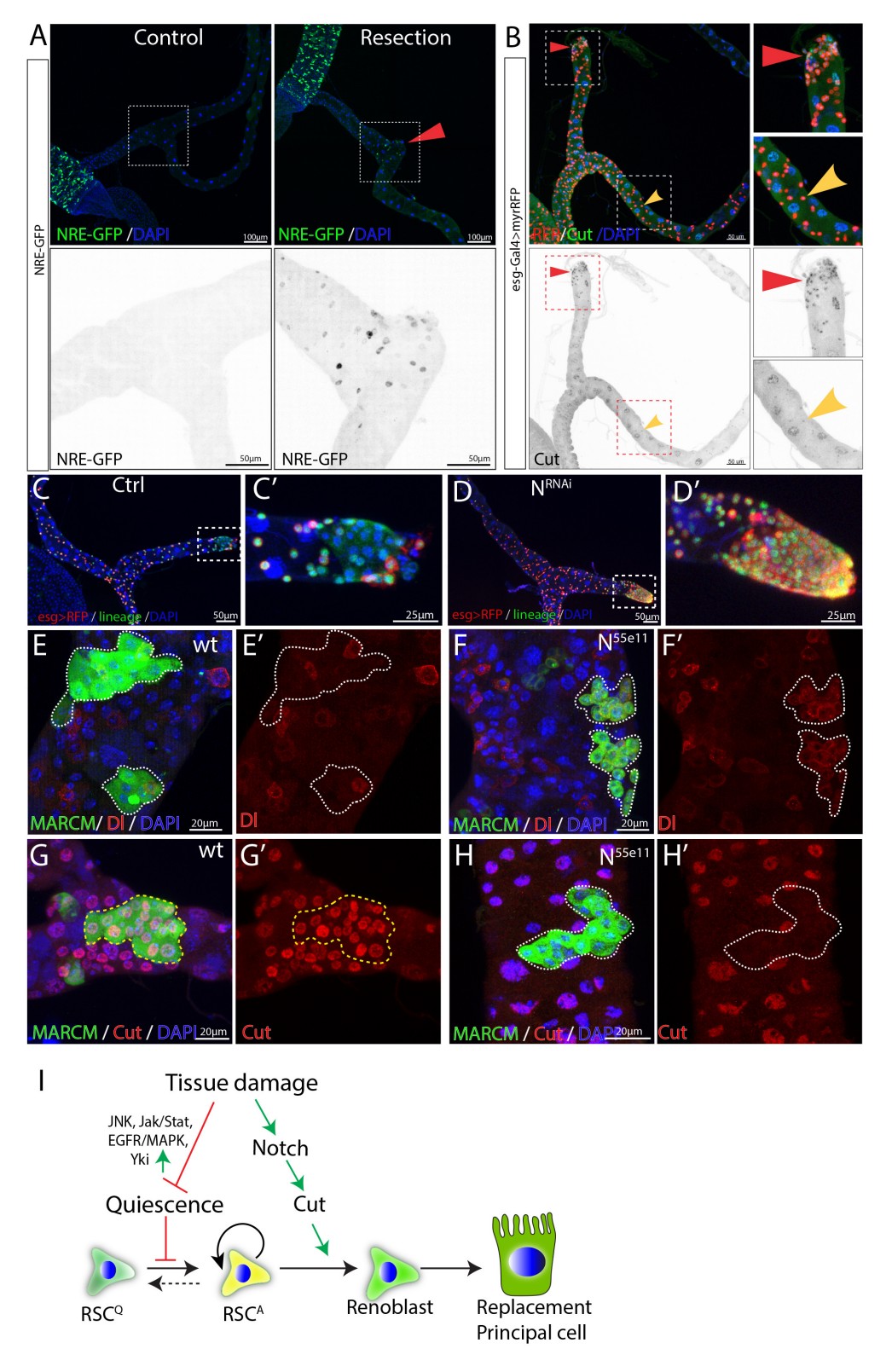

**Figure 4.** Damage activates Notch signaling which in turn regulates differentiation of RSCs. (**A**) Expression of NRE-GFP was barely detectable in the SCZ under normal conditions. In contrast, surgical resection of a Malpighian tubule induced the expression of NRE-GFP near the surgical site (indicated by red triangle). (**B**) Expression of Cut is upregulated in the esg >RFP+ cells around the surgical site (red triangles) compared to esg >RFP+ cells in the
*Figure 4 continued on next page*

*Figure 4 continued*

intact Malpighian tubules (indicated by yellow arrowhead). (**C–D**) Lineage analysis of control (**C**) and Notch-depleted (**D**) RSCs following surgical resection of Malpighian tubule (See Materials and methods). Expression of N[RNAi] driven by *esg[ts]* for 14 days resulted in accumulation of RSCs near the surgical site. (**E–H**) MARCM clones showing that *N[55e11]* mutant cells (**F,H**) express Dl and fail to polyploidize and turn on the PC differentiation marker Cut. Control in (**E**) and (**G**). MARCM clones were induced in animals fed on Allopurinol food for 7–10 days prior to heatshock. (**I**) Model for quiescent RSC-mediated homeostasis maintenance of the ureter and lower tubules. In normal conditions, RSCs are in a quiescent state. Damage to the ureter and lower tubules promotes the exit from quiescence through upregulation of multiple pathways including JNK, Jak/Stat and EGFR/MAPK etc. Meanwhile, damage also upregulates Notch signaling pathway which in turn upregulates Cut to ensure that activated RSCs (RSC[A]) only differentiate into replacement principal cells.

The online version of this article includes the following figure supplement(s) for figure 4:

**Figure supplement 1.** Forced expression of NICD or *ct* is sufficient to drive RSC differentiation.

---

GFP were generated using the MARCM system in animals that had been fed with the nephrotoxic drug allopurinol (*Lee and Luo, 1999*). Wild type RSCs clones typically consisted of multiple Cut[+] replacement principal cells along with one or more Dl[+] RSCs (*Figure 4E and G*). In contrast, *N[55e11]* mutant RSCs failed to upregulate Cut and polyploidize to form replacement principal cells. Instead they continued to produce cells expressing Dl (*Figure 4F and H*). These observations suggest that upon damage, RSCs can divide asymmetrically and activate the Notch pathway in one daughter cell, which turns on Cut and differentiates into an RB, a precursor that subsequently differentiates into a principal cell.

Consistent with this model, forced expression of the Notch intracellular domain (NICD), a constitutively active form of Notch using *esg-Gal4[ts]* while tagging RSCs with esg >RFP caused depletion of esg >RFP[+] cells after 14 days shifted to 29°C. The depletion of esg >RFP[+] cells was not due to cell death, as they still expressed the lineage marker (*Figure 4—figure supplement 1A and B*). *ct* is a well-known Notch target (*Sun and Deng, 2005*) that has been suggested to play an important role in renal differentiation and to be expressed in RSCs at low levels (*Xu et al., 2018*). We noticed that Cut expression was upregulated in regenerating cells in the lower tubule after surgery, suggesting that its upregulation might promote replacement PC production (*Figure 4B*). Indeed, expression of *UAS-ct* with *esg-Gal4[ts]* recapitulated the NICD-caused depletion of esg[+] cells (*Figure 4—figure supplement 1C*). In addition, forced expression of *ct* or forced expression of NICD in RSCs caused them to differentiate towards small replacement PCs expressing Alp4 (*Figure 4—figure supplement 1D–F*). Conversely, RNAi-mediated knockdown of ct inhibited the differentiation of injury-activated RSCs towards Alp4[+] replacement PCs (Figure 4-figure supplement 1G-I), indicating ct is required for PC differentiation. Thus as in ISCs, Notch signaling is required and sufficient for RSC differentiation. However, unlike differentiating midgut cells, PCs require the upregulation of *ct* in RSC daughters (*Figure 4I*).

## RSCs can switch between symmetric and asymmetric division after injury

The large increase in diploid renal cells that accompanies regeneration of multiple PCs raised the question of how multiple PC progenitors (RBs) arise. In particular, RBs could all be produced by repeated asymmetric divisions of RSCs, or they could be expanded by symmetric divisions of already formed RBs. RBs are Notch-induced and can be labeled by NRE-GFP. We first examined which cell types undergo mitotic division in the regenerating Malpighian tubules. Following expression of *rpr* and *hid* with *c507-Gal4[ts]* for 7 days, mitoses were readily detected. All phospho-Histone H3 (PH3)-marked mitotic cells were also positive for Dl and negative for NRE-GFP (*Figure 5A–C*). These data demonstrate that RSCs are the only cells that divide after injury in the adult Malpighian tubules.

Replacement cells could be produced using fewer divisions if RSCs divided symmetrically to expand in number, before or in concert with asymmetric divisions to produce new RBs. Previous studies have shown that ISCs can switch between symmetric and asymmetric modes to adapt to the physiological stimuli (*O'Brien et al., 2011*). We thus investigated the division mode of RSCs following acute injury elicited by surgery. Using the twin-spot MARCM system (*Yu et al., 2009*), we induced twin-spot MARCM clones 1 day after surgical damage to the Malpighian tubules. Twin-spot

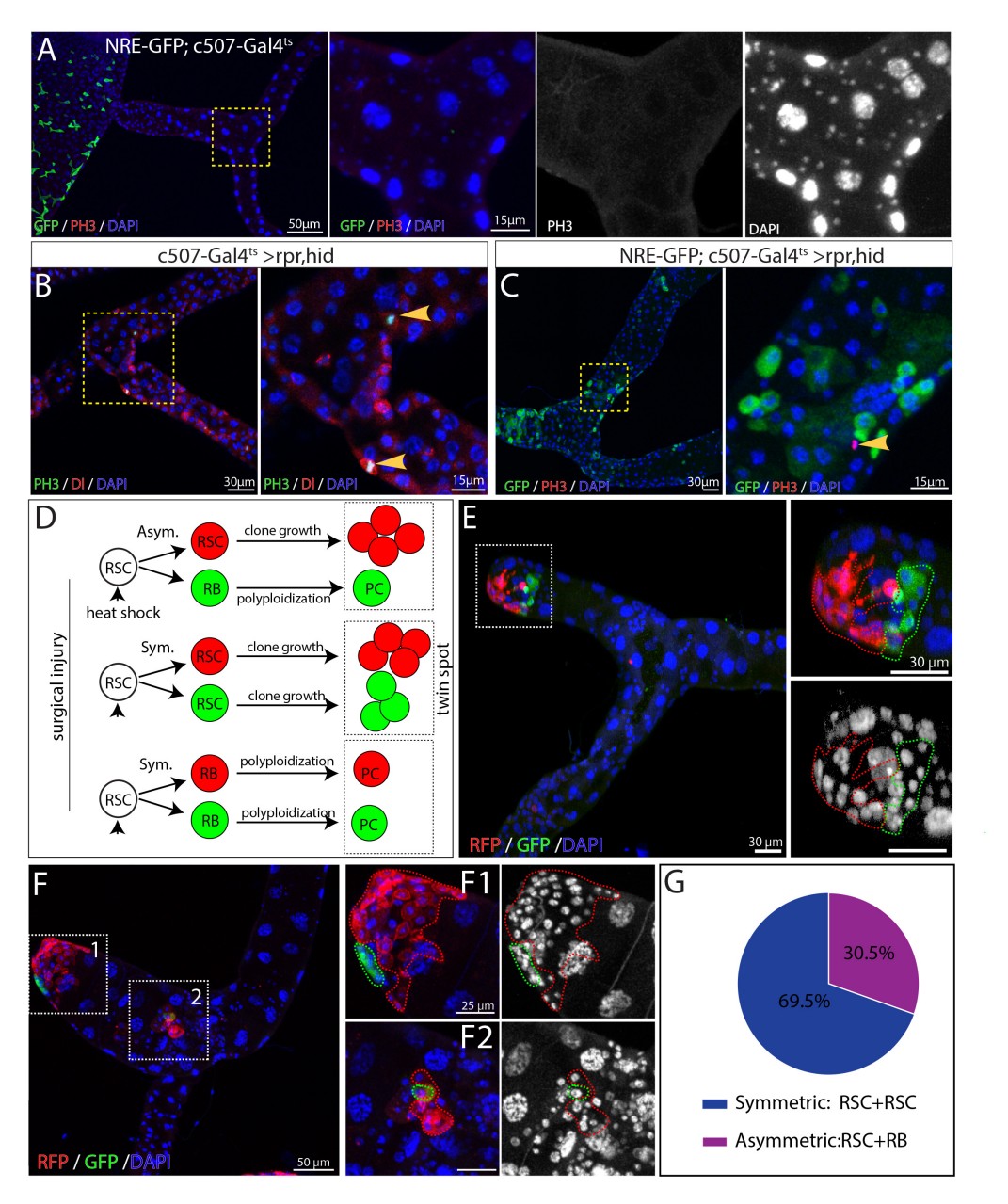

**Figure 5.** Activated RSCs can undergo both symmetric and asymmetric division upon damage. (**A–C**) Expression of rpr and hid driven by c507-Gal4^ts for 7 days at 29°C was used to ablate PCs in the SCZ. (**A**) NRE-GFP⁺ cells and mitotic cells were barely detectable in MTs of control animals (*NRE-GFP;c507-Gal4^ts*, 7 days at 29°C). (**B**) Mitotic cells (marked by PH3 staining, green) expressed the RSC marker Dl (red) in regenerating Malpighian tubules. (**C**) Mitotic cells (red) were negative for renoblast (RB) marker NRE-GFP (green) in regenerating Malpighian tubules. (**D–F**) Twin-spot MARCM system was used to generate RFP and GFP-labeled twin spots. One day before clone induction, animals with appropriate genotype were subjected to surgical resection of upper tubules to activate the quiescent RSCs. (**D**) Predicted twin-spot fates after asymmetric or symmetric division. (**E**) Example of an RSC/RSC pair of twin spots at day seven after clone induction. (**F**) Example showing an RSC/RSC of pair twin spots (**F1**) and an RSC/RB pair of twin spots (**F2**). (**G**) Observed percentage of symmetric division and asymmetric division.

MARCM system allowed us to label both daughter cells with permanent lineage markers GFP or RFP after heat-shock induced mitotic recombination. By examining the clone-forming capacity of the labeled daughter pairs, we inferred the individual fates of labeled daughter cells and thus determined whether they were produced by symmetric or asymmetric division of their mother stem cell

(*Figure 5D*). On day 4–7 after clone induction, we observed both symmetric lineages and asymmetric lineages in the SCZ. Approximately 70% of twin spots (n = 46) were derived from RSC/RSC pairs since both RFP-marked clone and GFP-marked clone had multiple cells (*Figure 5E–G*). Interestingly, all RSC/RSC pairs were observed at the surgical sites, where the replacement cells were most urgently required. In addition the RSC closer to the surgical site usually gave rise to more cells than its sibling RSC did (*Figure 5E and F*-F1). The RSC/RB pairs, which gave rise to approximately 30% of examined twin spots, were localized several PCs away from surgical sites (*Figure 5F and F2*). These results show that RSCs employ symmetric as well as asymmetric divisions to generate replacement PCs.

## Renal stem cells are normally quiescent

We examined the expression of the mitotic marker PH3 among RSCs labeled by esg >RFP in order to estimate the frequency of RSC division in the absence of injury. No mitotic cells were observed in wild-type Malpighian tubules (n = 47) under normal conditions (*Figure 6A and C*), in contrast to previous claims (*Singh et al., 2007*). However, 10.7 ± 5.1 (mean ± SD, n = 18) mitotic cells on average were seen per pair of Malpighian tubules after expression of esg >*upd1* in RSCs for 7 days. EdU incorporation over 2d was used as a more sensitive system to detect RSC division under normal conditions. EdU incorporation in midgut cells was widespread within this interval, reflecting the high stem cell activity in this tissue (*Figure 6D*). However, very few RSCs (~0.3%) were labeled in the same period (*Figure 6D and E*). After 4d of labeling, EdU incorporation could be detected in only about 1% of RSCs (*Figure 6E*). We conclude that RSCs are unlike ISCs and do not actively maintain renal cells in the absence of injury causing cell loss.

The Fly-FUCCI system was used to assess the cell cycle phase distribution of RSCs (*Zielke et al., 2014*). Fly-FUCCI is based on combination of fluorochrome-tagged degrons from the Cyclin B and E2F1 proteins, which are degraded during mitosis or S phase (*Figure 6—figure supplement 1A*), respectively. Expression of $UAS\text{-}CFP\text{-}E2F1_{1\text{-}230};UAS\text{-}Venus\text{-}CycB_{1\text{-}266}$ driven in RSCs by *esg-Gal4, UAS-myrRFP* showed that 42% of RSCs (n = 461 cells analyzed) expressed both $CFP\text{-}E2F1_{1\text{-}230}$ and $Venus\text{-}CycB_{1\text{-}266}$ and had a 4C DNA content, indicative of G2 phase. Only 3.2% of RSCs expressed $Venus\text{-}CycB_{1\text{-}266}$ but not $CFP\text{-}E2F1_{1\text{-}230}$, consistent with the low percentage of RSCs in S phase revealed by EdU incorporation. 9.8% of RSCs expressed $CFP\text{-}E2F1_{1\text{-}230}$ but not $Venus\text{-}CycB_{1\text{-}266}$, indicative of G1 phase. Interestingly, 47% of esg >RFP$^+$ RSCs expressed neither $CFP\text{-}E2F1_{1\text{-}230}$ nor $Venus\text{-}CycB_{1\text{-}266}$, suggesting these cells lack expression of cell cycle progression factors (*Figure 6F, Figure 6—figure supplement 1B*). In addition, the cells double negative for E2F1 and CycB have a 2C DNA content (*Figure 6—figure supplement 1C*), characteristics of G0 state (*Cheung and Rando, 2013*). Taken together, these data indicated that the vast majority of RSCs are arrested at G2 or G0 phase under non-stress conditions.

## Renal stem cells are unipotent

To further study the behavior of RSCs, we used the MARCM system to permanently mark individual dividing RSCs and their progeny via heat-shock induced mitotic recombination (*Lee and Luo, 1999*). Without heat shock, we observed very sparse clones in midgut and Malpighian tubules, showing that the MARCM system was associated with little background clone induction (*Figure 6G, Figure 6—figure supplement 2B*). After a low dose heat shock at 37°C, a substantial number of clones were induced in the midgut reflecting the active ISC division in this tissue, but clones were not detected among quiescent RSCs of the Malpighian tubules (*Figure 6H, Figure 6—figure supplement 2B*). However, 6 successive 1 hr heat shocks spaced 24 hr apart (6X hs) were found to induce RSC division (*Figure 6H'–I'*), either because such treatment damaged PCs and stimulated regeneration, or else over-rode the normal controls of RSC proliferation. In support of the notion that excessive heat stress stimulates proliferation of RSCs, the stress-responsive JNK signaling pathway was induced by 6X heat shock in Malpighian tubules (*Figure 6—figure supplement 2A*). In addition, clones induced by 6X heat shock contained more cells than the rare clones induced by 2X heat shock (*Figure 6—figure supplement 2B*).

The induced MARCM clones were confined to the SCZ (*Figure 6I*), and none were observed in the upper tubules (N = 1957 clones, n > 100 flies), indicating that they were generated by RSCs. We examined the cellular composition of these marked clones to probe the differentiation potential of

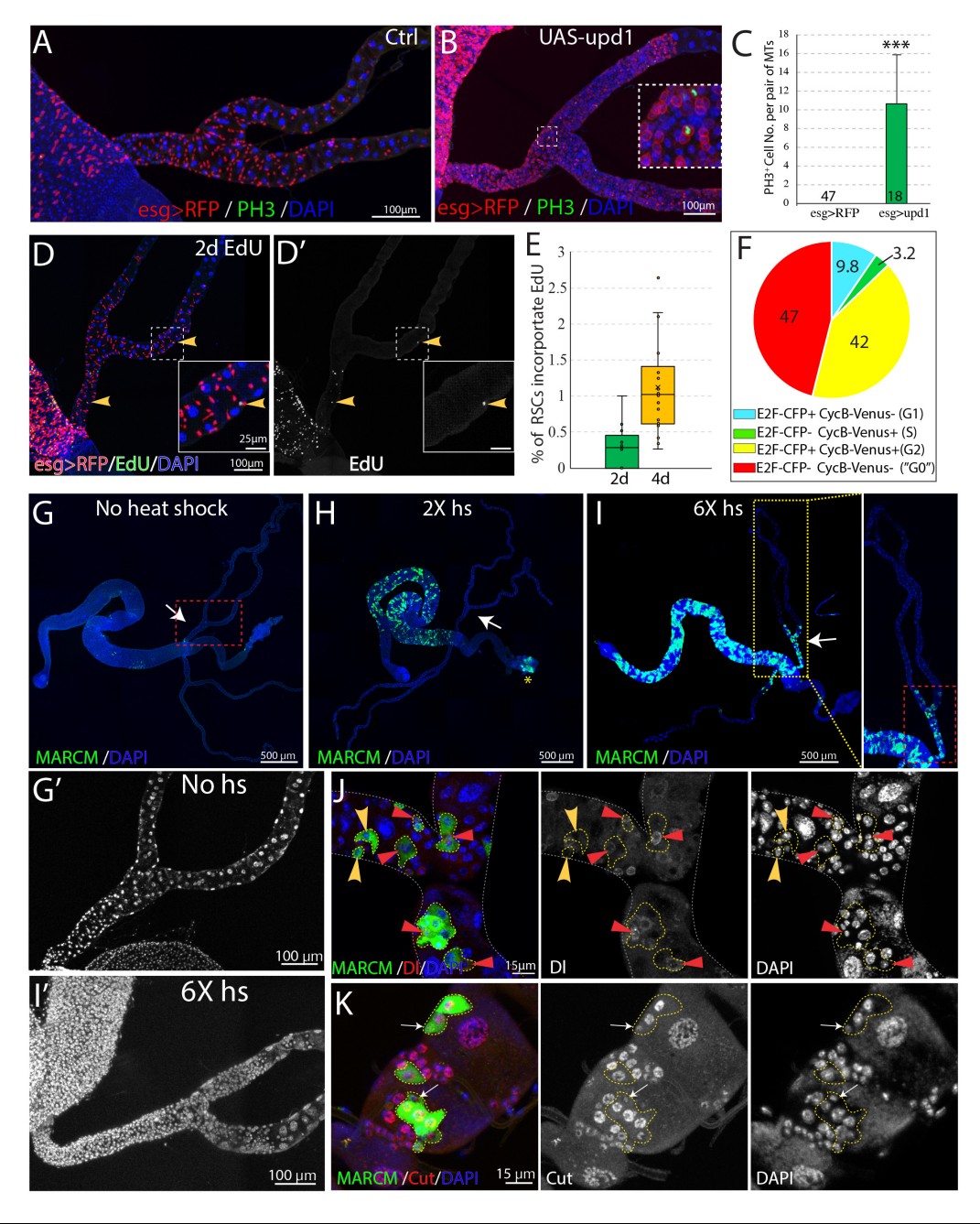

**Figure 6.** RSCs are normally quiescent. (**A and B**) Phospho-histone H3 (Ser10) (PH3) staining was done to indicate mitoses in *esg-Gal4$^{ts}$ > RFP* (**A**) and *esg-Gal4$^{ts}$ > RFP+UAS-upd1* (**B**) Malpighian tubules. After 6 days of expression of Upd1, mitoses were readily present. Inset: enlarged image of the dashed box showing PH3$^+$ cells are esg >RFP$^+$ RSCs. (**C**) Quantification of RSCs proliferation induced by expression of *UAS-upd1*. Data represent means ± SD, *** denotes Student's t test p<0.001. (**D**) Representative image of EdU labeling after 2d in wild type Malpighian tubules. Yellow arrowheads denote esg >RFP+ RSCs labeled with EdU. Insets are enlarged images of dashed boxes. (**E**) Quantified percentage of EdU-positive RSCs after continuous EdU labeling for 2 days and 4 days, n > 10 female flies. (**F**) Cell cycle phase distribution of RSCs as assessed by FUCCI. (**G–J**) MARCM clones induced by different heat shock dose. Compared to no heat shock control (**G**), there was a significant increase in the GFP marked clones in the midgut but not Malpighian tubules after 2X heat shocks (**H**); the yellow asterisk denotes the autofluorescence from fecal contents in the rectum. (**I**) Induced clones were seen in the SCZ with 6X heat shocks. As expected, labeling was restricted to the ureter and lower tubules, where RSCs are located. (**G′ and I′**) Enlarged DAPI channels of the SCZ (dashed red box) corresponding to G and I, respectively. (**J**) Representative

*Figure 6 continued on next page*

*Figure 6 continued*

image showing both single cell transient clones (outlined in dashes; the single cell is denoted by yellow arrowhead) and multi-cell RSC clones (outlined in dashes), containing a single Dl$^+$ RSC (denoted by red triangle). No Dl$^+$ cells were detected in the single-cell transient clones. (K) Representative image of GFP marked clones in the lower tubules showing the differential expression levels of Cut in different cell types in RSC clones (outlined by yellow dashes). Cut is expressed at a low level in a single 2 C cell in each clone that is presumably an RSC (white arrows). Downstream cells that are often of higher ploidy show higher Cut expression. For (C), (E) and (F), see also *Figure 6—source data 1*.

The online version of this article includes the following source data and figure supplement(s) for figure 6:

**Source data 1.** Source data for *Figure 6E-F*.
**Figure supplement 1.** esg+ cells are largely arrested at G0 and G2 phases in normal condition.
**Figure supplement 2.** Excessive heat stress activates JNK pathway and promotes clone labeling and growth.

RSCs. 7 days after clone induction, both single-cell clones and multi-cell clones were detected. Single-cell clones comprise only one polyploid cell negative for Dl, as expected if the RB was marked following an asymmetric RSC division. In addition, multi-cell RSC clones were observed that consisted of one Dl$^+$2 C cell and several Dl$^-$ polyploid cells (*Figure 6J*). Staining of Cut showed that the RSC clones comprise a 2 C cell with low level expression of Cut and 2–4 polyploid cells expressing higher level of Cut that represent newly completed PCs derived from the RSC (*Figure 6K*). Notably, new principal cells derived from RSCs are 8–16C and never reach the ploidy of preexisting principal cells (64–128C).

## Renal stem cells respond to injury caused by kidney stones

While protection from extreme heat stress provides one possible reason *Drosophila* kidneys contain RSCs, we also investigated another potential problem, cell loss caused by kidney stones (*Figure 7*, *Figure 7—figure supplement 1*). Xanthine stones in *Drosophila* Malpighian tubules (analogous to human hereditary xanthinuria) can be reproducibly generated by mutating the *rosy* (*ry*) gene, encoding xanthine dehydrogenase (XDH) (*Mitchell and Glassman, 1959*; *Bonse, 1967*; *Figure 7—figure supplement 1A–C*), or by feeding adults the XDH inhibitor allopurinol (*Chi et al., 2015*; *Figure 7—figure supplement 1D*). Although control flies did not show Malpighian tubule stones even after 14d, 80% or more of *ry* mutant adults contained stones in the SCZ within 1d of eclosion, and the stones grew in size over the next 4 weeks (*Figure 7—figure supplement 1C*). Stones appeared first in the ureter and lower tubules, but could eventually be found in more distal regions. Similar effects were observed in flies treated with allopurinol (*Figure 7—figure supplement 1D*). *ry* mutant Malpighian tubules with stones suffered a loss of nearby PCs due to detachment from the tubule epithelium and movement into the lumen, a process that frequently started near the junction of the lower segment with the ureter (*Figure 7A–7D*, *Figure 7—figure supplement 1B*). In their place, many new small cells appeared, some of which showed evidence of polyploidization (*Figure 7E*). We noticed that the number of cells positively correlated with the size of stones in the SCZ (*Figure 7—figure supplement 1E and F*), demonstrating that it is stone-induced damage activating RSCs to proliferate and replenish the SCZ.

Consistent with this notion, transcriptome profiling by mRNA-seq of dissected SCZ from *ry$^{506}$* mutants carrying stones revealed that many genes associated with mitotic cycling and RSC activation were increased relative to wild type (*Figure 7—figure supplement 2A*). In consonance with surgical injury studies, activation of the JNK, Jak/Stat, EGFR, Yki and Notch signaling was induced by stones as revealed by both mRNA-seq and reporter assays (*Figure 7—figure supplement 2*).

Likewise, allopurinol treatment of wild type flies greatly increased the number of total cells, as well as *esg$^+$* cells in the SCZ that incorporate EdU (*Figure 7—figure supplement 1G–H*). *esg-Gal4$^{ts}$* directed Flp-out lacZ marked lineage tracing showed that the increased cells were derived from *esg$^+$* cells (data not shown). The distribution of cell cycle stages of RSCs in stone-carrying animals was different from those of RSCs in control animals (*Figure 7—figure supplement 1I*). The percentage of RSCs at G0 decreased from 47% to 4.6% (n = 624 cells). In addition, the number of mitotic clones was significantly increased after Allopurinol treatment (*Figure 7—figure supplement 1J–L*). Molecular analysis of clones validated that RSCs give rise to daughter cells that differentiate into new principal cells as indicated by the high level expression of Cut (*Figure 4G*). However, clonal

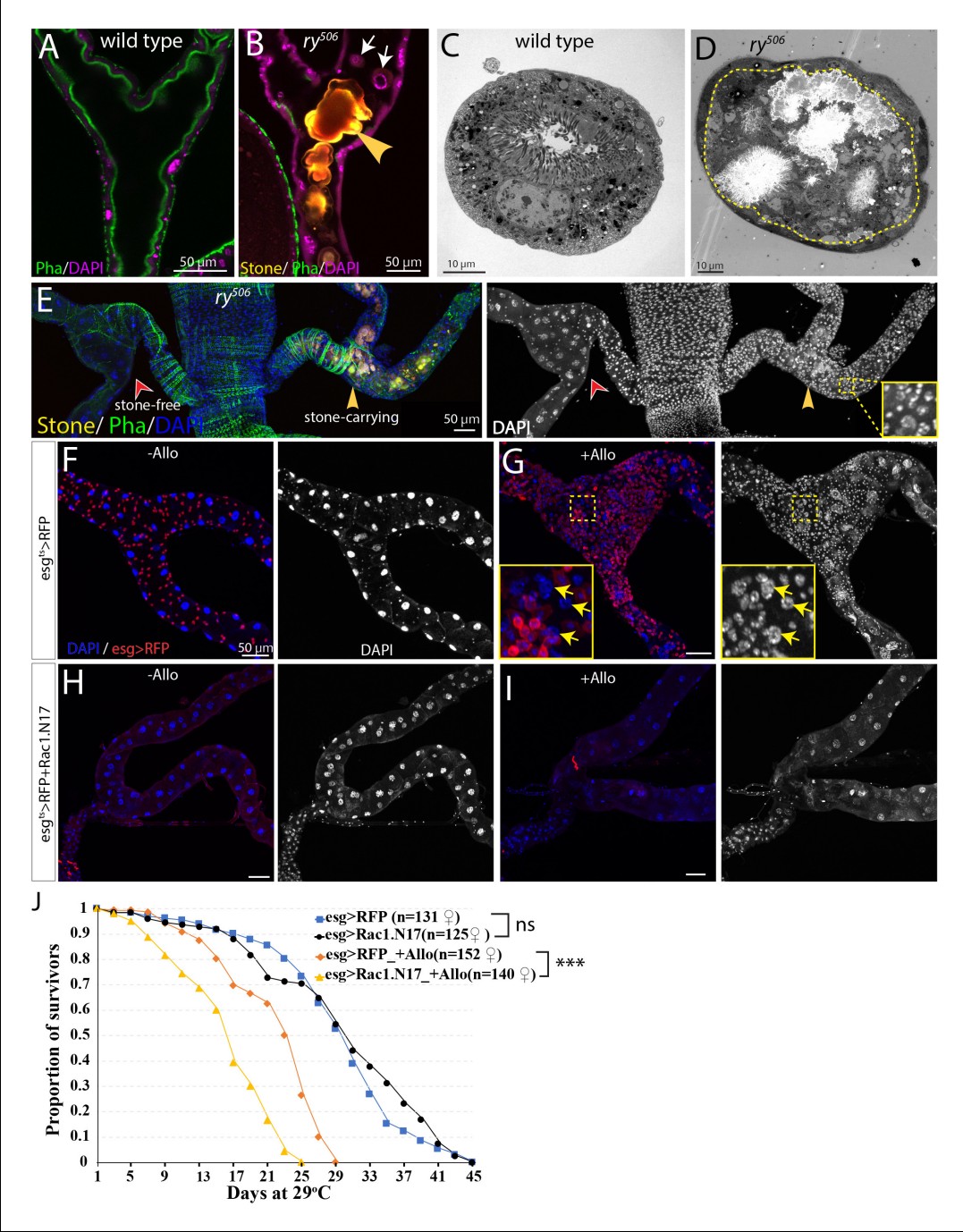

**Figure 7.** RSCs respond to stone formation. (**A**) Longitudinal optical section view of the ureter and lower tubules from a wild-type (Oregon-R) animal. The surrounding muscles and brush borders of principal cells are marked by phalloidin staining (green). (**B**) Longitudinal section view of the ureter and lower tubules from $ry^{506}$ mutant bearing stones (yellow arrowhead) in tubule lumen. Note that detached principal cells (white arrows) were found in tubule lumen. DAPI channel was consciously over-exposed to reveal the nuclei of the dying principal cells in the lumen. (**C–D**) Representative electron micrographs showing stones and cell debris were seen in the $ry^{506}$ lower tubule (**D**) but not in the Oregon-R tubule (**C**). (**E**) Immunofluorescence image showing that supernumerary tubule cells were present in the stones-carrying tubules (yellow arrowheads) compared to the stone-free tubules (red arrowheads) in the same 14-day-old $ry^{506}$ fly. Shown on the right is the greyscale DAPI channel with an inset (outlined with yellow square) showing an closeup view of DAPI staining. Note the polyploid cells are produced in the damaged tubules. (**F–J**) RSCs are essential for ureter and lower tubule repair. Representative images showing that ureter and lower tubules from $esg^{ts} > RFP$ flies cultured on normal food (**F**) and allopurinol food (**G**) as well as those from

*Figure 7 continued on next page*

*Figure 7 continued*

*esg^{ts}* > *RFP+Rac1.N17* flies cultured on normal food (**H**) and allopurinol food (**I**) for 14 days after eclosion. Insets (outlined by yellow boxes) showing the presence of polyploid cells (yellow arrows) are shown in (**G**). Flies were shifted to 29°C since late L3 to induce the expression of Rac1.N17. Expression of *Rac1.N17* inhibited formation of RSCs in the ureter and lower tubules and completely blocked allopurinol induced production of supernumerary tubule cells. (**J**) RSCs-depleted flies showed a shortened lifespan compared to control flies when reared on allopurinol food. ns denotes log-rank test p>0.05, *** denotes log-rank test p<0.001. See also **Figure 7—source data 1**.

The online version of this article includes the following source data and figure supplement(s) for figure 7:

**Source data 1.** Source data for **Figure 7J**.
**Figure supplement 1.** RSCs respond to stone formation.
**Figure supplement 2.** Stone-induced damage causes activation of multiple proliferation pathways.

---

marking again showed that the RSC-generated cells did not replace distal PCs or migrate from the lower to the more distal regions of the Malpighian tubule (**Figure 7—figure supplement 1M–N**).

If *esg*⁺ cells are prevented from migrating into the SCZ by expressing a dominant negative form of Rac1 (Rac1.N17) during pupal development such that adults lack RSCs (**Figure 7H**, (**Takashima et al., 2013**), then few new small cells or polyploid PCs were produced after allopurinol treatment (**Figure 7I**). The lifetime of such RSC-depleted animals on allopurinol was significantly reduced compared to treated animals that retained their RSCs (**Figure 7J**). These observations show that RSCs are required to generate new replacement PCs in response to damage and that RSC-mediated regeneration is apparently advantageous for survival under the conditions of kidney stress.

## Discussion

### Reserved regenerative cells distribute to the regions of tissues that are targeted by injury

Adult stem cells are widely known to exist as rare cell populations in tissues and organs. Here we show that adult *Drosophila* RSCs are an exception. RSCs significantly outnumber the principal cells that they maintain, yet do not divide unless a neighboring principal cell is lost. Interestingly, these large RSC populations are confined to the ureter and lower tubules, and are not found throughout the Malpighian tubes. We observed that xanthine stones often start to form in the ureter and lower tubules, and frequently damage these regions, suggesting that the deployment of RSCs in the SCZ is driven by evolutionary selection pressure for regenerative capacity in regions where stones are most likely.

It has long been known that some parts of mammalian kidneys such as proximal tubule undergo cellular regeneration after acute kidney injury (AKI) (**Witzgall et al., 1994**). The S3 segment of proximal tubule is often damaged in cases of acute kidney injury elicited by ischemia, sepsis, trauma or other mechanisms (**Lazzeri et al., 2019**). However, it has been controversial as to whether tubular stem cells or progenitor cells contribute to regeneration. The long-standing idea is that the repair and regeneration of nephron tubules are predominantly based on the dedifferentiation of surviving tubular cells, which in turn divide to replace lost cells (**Humphreys et al., 2008**; **Humphreys et al., 2011**; **Kusaba et al., 2014**; **Kumar et al., 2015**). However, tissue-resident tubular progenitor cells, which are more resistant to death, have recently been characterized and demonstrated to play an important role in murine tubular regeneration (**Angelotti et al., 2012**; **Rinkevich et al., 2014**; **Kang et al., 2016**; **Lazzeri et al., 2018**). Thus, it is plausible that both *Drosophila* and mice employ reserved progenitor cells (RSCs in *Drosophila* or tubular progenitor cells in mice) to replenish specific cells in renal regions that are vulnerable to injury. We propose that RSC-like quiescent stem cells of limited potency exist in tissue regions that are preferential targets of injury, in multiple tissues with little or no normal cell turnover.

### G0 and G2 arrest of adult *Drosophila* renal stem cells

Active stem cells such as intestinal stem cells and germline stem cells have been extensively studied in *Drosophila*. However, studies on *Drosophila* quiescent stem cells are just emerging

(*Chaturvedi et al., 2017*; *Boukhatmi and Bray, 2018*; *Otsuki and Brand, 2018*). Although quiescent stem cells have been identified in many tissues, the characteristics of quiescent stem cells are still not well understood since they represent a very rare population in most tissues (*Li and Clevers, 2010*; *Cheung and Rando, 2013*). Adult *Drosophila* renal stem cells are abundant and comprise approximately 70% of total cells in the ureter and the tubule. Thus, *Drosophila* RSCs represent a favorable model to further understand quiescent stem cells.

Quiescent stem cells are commonly believed to arrest in G0/G1. RSCs, in contrast, were found to arrest in G0 (47%) or G2 (42%) (*Figure 6F*). Although it might appear to be counterintuitive, accumulating evidence shows that quiescent stem cells frequently arrest in G2. For instance, the majority of *Drosophila* neural stem cells become quiescent and are arrested in G2 phase for 24 hr after embryogenesis (*Otsuki and Brand, 2018*). G2 arrest is essential to maintain postnatal muscle stem cells in zebrafish (*Nguyen et al., 2017*). We propose that the bimodal arrest observed for RSCs is advantageous since G2 RSCs would be well positioned to respond promptly to injury, while G0 RSCs could serve as long term reservoirs. Since RSCs decline in number over time, it would be interesting to determine whether there is a selective change in the distribution of cell cycle phases.

## Why are adult-derived PCs lower in ploidy than embryo-derived PCs?

It is widely assumed that during regeneration, stem cells generate daughters that differentiate into identical replacement tissue, but RSCs generate replacement PCs of substantially lower in ploidy compared to preexisting PCs carried from the larva, in most of the SCZ. The final ploidy levels of both ISC and RSC progenitors are not entirely invariant, but usually remain within narrow limits. Daughter cells are sent on a course of differentiation via Notch signaling, and endocycles continue until a final tissue size is sensed. Likewise, in the *Drosophila* follicle cell lineage, mitotic divisions downstream from the stem cell continue until a mitotic to endocycle transition signals the onset of differentiation (*Deng et al., 2001*; *Sun and Deng, 2005*; *Ohlstein and Spradling, 2007*). If the number of cells differs from normal or changes during growth, the remaining cells will proliferate for a greater or lesser number of endocycles until the target tissue size is achieved (*Deng et al., 2001*; *Edgar et al., 2014*; *Ovrebo and Edgar, 2018*). Adult PCs are probably lower ploidy because the number of progenitor cells is relatively large so that many RBs are rapidly generated relative to the number of lost PCs. This arrangement speeds regeneration since it takes a smaller number of endocycles to achieve the starting tissue mass using a larger number of progenitors. In contrast, in the embryo when the Malpighian tubules initially form, PCs are established and start to endoreduplicate when the tissue is very small. Thereafter, growth of larval Malpighian tubules is completely driven by the continued endoreduplication rather than cell division.

RSCs may also generate replacement PCs of smaller size, because the enormous adult PCs carried over from the larva are less adapted to an adult organ than the smaller cells typical for adult digestive tissues. The failure to replace the larval tubules during pupal development may be a physiological necessity for maintaining ionic and water balance during pupal development. Thus, repair to smaller size PCs during adulthood might provide a functional upgrade that completes the rebuilding of the tissue along the lines other tissues undergo during pupal development.

## Modeling kidney stone disease in *Drosophila*

One common form of kidney diseases is kidney stone formation, whose incidence and prevalence have been increasing over the past several decades worldwide (*Romero et al., 2010*). While non-obstructing stones do not cause significant damages apart from hematuria, obstructing stones can elicit both acute as well as irreversible chronic kidney injury (*Coe et al., 2005*; *Rule et al., 2011*). Much remains to be learned about the cellular and molecular mechanisms underlying kidney stone formation. Type I Hereditary Xanthinuria in humans, which is caused by mutation of *xanthine dehydrogenase* (aka *xanthine oxidoreductase*, *XDH/XOR*), can damage kidney and even lead to renal failure (*Akıncı et al., 2013*). Our data show that xanthine stones can cause sloughing of principal cells and shorten the lifespan of *Drosophila*. Additionally, preventing RSC-mediated regeneration of the SCZ in the presence of allopurinol-induced kidney stones shortens the lifespan, although we cannot exclude effects on the midgut of expressing *Rac1.N17* in esg+ ISCs/RSCs. Our data demonstrate that *Drosophila* can be used to model kidney stone formation and help understand stone-induced

damage and repair. Continued work on the formation and RSC-mediated repair of *Drosophila* stones is likely to provide new information relevant to the prevention and treatment of kidney stones.

## Materials and methods

### Key resource table

See *Supplementary file 1*, Key resource table.

### *Drosophila* and husbandry

The *Drosophila* stocks used are listed in the resource table. Flies were reared on normal cornmeal molasses food and maintained at room temperature (23–25°C) unless otherwise specified.

### Detailed genotypes for each figure

*Figure 1* (C and D) *esg-Gal4, UAS-myrRFP*

*Figure 2* (D and E) *esg-Gal4, UAS-myrRFP* (F) *esg-Gal4, UAS-myrRFP/NRE-GFP* (H) *esg-Gal4, UAS-myrRFP/ Df31-GFP* (I) *cad-Gal4/UAS-Redstinger*

*Figure 3* (A) *tub-Gal80$^{ts}$/+; c507-Gal4/+* (7d at 29°C) (B) *UAS-rpr,UAS-hid/+; tub-Gal80$^{ts}$/+; c507-Gal4/+* (7d at 29°C) (C) *UAS-rpr,UAS-hid/+; tub-Gal80$^{ts}$/+; c507-Gal4/+* (7d at 29°C, then shifted to 18°C for 21d) (G–I) *esg-Gal4, UAS-myrRFP*

*Figure 4* (A) *NRE-GFP* (B) *esg-Gal4, UAS-myrRFP* (C) *esg-Gal4, UAS-myrRFP, UAS-flp/Act5C>-STOP > lacZ ; tub-Gal80$^{ts}$/+* (D) *esg-Gal4, UAS-myrRFP, UAS-flp/Act5C>STOP > lacZ ; tub-Gal80$^{ts}$/UAS N-RNAi* (E and G) *hs-flp, FRT$^{19A}$, tub-Gal80/FRT$^{19A}$; AyGAL4, UAS-GFP /+* (F and H) *hs-flp, FRT$^{19A}$, tub-Gal80/FRT$^{19A}$ N$^{55e11}$; AyGAL4, UAS-GFP /+*

*Figure 5* (A) *tub-Gal80$^{ts}$/NRE-GFP; c507-Gal4/+* (B) *UAS-rpr,UAS-hid/+; tub-Gal80$^{ts}$/+; c507-Gal4/+* (C) *UAS-rpr,UAS-hid/+; tub-Gal80$^{ts}$/NRE-GFP; c507-Gal4/+* (E and F) *hs-flp/+; UAS-rCD2.RFP,UAS-GFPi, FRT$^{40A}$/UAS-rCD8.GFP,UAS-rCD2i,FRT$^{40A}$; tub-Gal4/+*

*Figure 6* (A) *esg-Gal4, UAS-myrRFP/+; tub-Gal80$^{ts}$/+* (6d at 29°C) (B) *esg-Gal4, UAS-myrRFP/ UAS-upd1; tub-Gal80$^{ts}$/+* (6d at 29°C) (D) *esg-Gal4, UAS-myrRFP/+* (G–K) *UAS-cd8GFP, hsFlp/+; FRT$^{42D}$ tub-Gal80/FRT$^{42D}$; tub-Gal4/+*

*Figure 7* (A and C) *Oregon-R* (B,D and E) *ry$^{506}$ -/-* (F and G) *esg-Gal4, UAS-myrRFP/+; tub-Gal80$^{ts}$/+* (shift to 29°C in the late L3 phase) (H and I) *esg-Gal4, UAS-myrRFP/UAS-Rac1.N17; tub-Gal80$^{ts}$/+* (shift to 29°C in the late L3 phase)

*Figure 1—figure supplement 1* (B) *31E09-Gal4/UAS -GFP* (C) *Pvr-GFP* (D) *UAS-myrRFP/+; c507-Gal4/+* (E) *Uro-Gal4/UAS-myrRFP* (F) *tsh-Gal4/UAS-myrRFP*

*Figure 3—figure supplement 1* (A) *tub-Gal80$^{ts}$/+; c507-Gal4, UAS-RFP /+* (B) *UAS-rpr,UAS-hid/+; tub-Gal80$^{ts}$/ +; c507-Gal4,UAS-RFP/+* (7d at 29°C, then shifted to 18°C for 21d) (C) *tsh-lacZ/+; c507-Gal4,UAS-RFP/+* (D) *UAS-rpr,UAS-hid/+; tub-Gal80$^{ts}$/tsh lacZ; c507-Gal4,UAS-RFP/+* (7d at 29°C, then shifted to 18°C for 21d) (F and G) *esg-Gal4, UAS-myrRFP, UAS-flp/Act5C>STOP > lacZ ; tub-Gal80$^{ts}$/+* (21 day at 29°C after surgery)

*Figure 3—figure supplement 2* (A) *tsh-Gal4,UAS-myrRFP/tsh-lacZ* (B) *tsh-Gal4,UAS-myrRFP/tsh-lacZ; tub-Gal80$^{ts}$/+* (ctrl) (C) *UAS-rpr,UAS-hid/+; tsh-Gal4,UAS-myrRFP/tsh-lacZ; tub-Gal80$^{ts}$/+* (7d at 29°C) (D) *UAS-rpr,UAS-hid/+; tsh-Gal4,UAS-myrRFP/tsh-lacZ; tub-Gal80$^{ts}$/+* (7d at 29°C, then shifted to 18°C for 21d) (F) *Uro-Gal4,UAS-RFP/+; tub-Gal80$^{ts}$/+* (7d at 29°C) (G) *UAS-rpr,UAS-hid/ +; Uro-Gal4,UAS-RFP/+; tub-Gal80$^{ts}$/+* (7d at 29°C, then shifted to 18°C for 21d)

*Figure 3—figure supplement 3* (A) *puc$^{E69}$-Gal4,UAS-GFP/+* (B) *10X stat92E-GFP* (C) *esg-Gal4, UAS-myrRFP/+* (D) *Diap1-lacZ /+*

*Figure 4—figure supplement 1* (A) *esg-Gal4, UAS-myrRFP, UAS-flp/Act5C>STOP > lacZ ; tub-Gal80$^{ts}$/+* (B) *esg-Gal4, UAS-myrRFP, UAS-flp/UAS- NICD; tub-Gal80$^{ts}$/Act5C>STOP > lacZ* (C) *esg-Gal4, UAS-myrRFP, UAS-flp/UAS- ct; tub-Gal80$^{ts}$/Act5C>STOP > lacZ* (D) *esg-Gal4,UAS-myrRFP/+; tub-Gal80$^{ts}$/Alp4-lacZ* (E) *esg-Gal4,UAS-myrRFP/UAS-NICD; tub-Gal80$^{ts}$/Alp4-lacZ* (F) *esg-Gal4,UAS-myrRFP/UAS-ct; tub-Gal80$^{ts}$/Alp4-lacZ* (G) *esg-Gal4,UAS-myrRFP/tub-Gal80$^{ts}$; Alp4-lacZ/UAS-ct-RNAi* (7d at 29°C, reared on regular food) (H) *esg-Gal4,UAS-myrRFP/tub-Gal80$^{ts}$; Alp4-lacZ/+* (7d at 29°C, reared on Allopurinol food) (I) *esg-Gal4,UAS-myrRFP/tub-Gal80$^{ts}$; Alp4-lacZ/UAS-ct-RNAi* (7d at 29°C, reared on Allopurinol food)

*Figure 6—figure supplement 1* (B and C) *esg-Gal4,UAS-myrRFP/UAS CFP.E2f1.1–230; UAS-Venus.CycB.1–266/+*

*Figure 6—figure supplement 2* (A) *puc$^{E69}$-Gal4,UAS-GFP/+*

*Figure 7—figure supplement 1* (B and E) *ry*$^{506}$ -/- (G and H) *esg-Gal4,UAS-myrRFP/+* (J,K,M) *hs-flp/+; UAS-rCD2.RFP,UAS-GFPi, FRT*$^{40A}$*/UAS-rCD8.GFP,UAS-rCD2i,FRT*$^{40A}$*; tub-Gal4/+*

*Figure 7—figure supplement 2* (B) *puc*$^{E69}$*-Gal4,UAS-GFP/+* (C) *10Xstat92E-GFP* (D) *Diap1-lacZ* (E) *ex-lacZ* (F) *NRE-GFP/+; ry*$^{506}$ *+/-* (G) *NRE-GFP/+; ry*$^{506}$ *-/-.*

## Immunostaining and microscopy

Malpighian tubules were dissected in Grace's insect buffer and fixed in 4% paraformaldehyde on a nutator at room temperature for 20 min. Sample were then washed 3 times in PBT (1XPBS + 0.1% TritonX-100) and blocked in PBT plus 5% NGS for 1 hr followed by primary antibodies incubation overnight at 4°C. Secondary antibodies were incubated for 2–3 hr at room temperature or overnight at 4°C. To minimize the autofluorescence elicited by ureter stones, Malpighian tubules bearing ureter stones were dissected and fixed in 800 ul fixative solution (100 ul 16% EM-grade paraformaldehyde + 300 ul Grace's insect buffer + 400 ul n-heptane) on a nutator at room temperature for 20 min. After fixation, the aqueous phase was removed and 400 ul 100% methanol was added, followed by vigorous hand shaking for 30 s. Samples were subsequently washed with 1 ml of 100% and 50% methanol for 5 min respectively. Samples were then stained as described above. Samples were imaged with a Leica TCS SP5 or Leica TCS SP8 confocal microscope.

## Electron microscopy

Malpighian tubules were dissected in Grace's Buffer and the gut and upper tubules were carefully removed. Samples were fixed for 1 hr in fixative solution (3% glutaraldehyde + 1% formaldehyde + 0.1 M cacodylate buffer + 2 mM CaCl$_2$, pH = 7.4). Following rinse with 0.1 M cacodylate buffer samples were embedded in agarose at 50°C, washed with cacodylate buffer three times for10 minutes each time. Then the samples were further fixed in 1% OsO$_4$ + 1% KFeCN in cacodylate buffer for 45 min. The samples were then processed and embedded as previously described (*Marianes and Spradling, 2013*). Electron microscopy images were captured with a Phillips Tecnai 12 microscope.

## Allopurinol feeding

Allopurinol stock solution (300 mM) was made by dissolving 0.41 g of Allopurinol (Sigma-Aldrich, cat# A8003-25g) in 10 ml of 1M sodium hydroxide. Normal cornmeal molasses food was melted and supplemented with 1/100 vol of allopurinol stock solution. After mixing well, the allopurinol food was poured into individual vials. Female flies with appropriate genotypes were cultured on allopurinol food as indicated in figure legends.

## Heat shock scheme for mosaic analysis

MARCM systems were used to generate mitotic clones (*Lee and Luo, 1999*). Flies with appropriate genotypes were put in empty vials augmented with yeast paste on the wall and heat-shocked in a 37°C water bath once to multiple times (60 min each time), with an interval of 24 hr between heat shocks. To assay the proliferative activity of RSCs upon injury caused by ureter stones, newly eclosed female flies were reared on cornmeal-molasses food containing 3 mM Allopurinol for 7 days prior to heat shock. After 2X heat shocks, flies were transferred to Allopurinol food and transferred every 2–3 days.

For twin-spot MARCM (*Yu et al., 2009*), animals were subjected to surgical injury. One of the anterior pair of Malpighian tubules was severed in the SCZ. After recovery for 1 day, animals were heat-shocked in a 37°C water bath for 60 min. The resultant RFP-marked and GFP-marked twin-spots were scored as 'symmetric' if both clones contained multiple cells at day 5–7 after clone induction. Twin spots were scored as 'asymmetric' if they consisted of one polyploid cell in one color and multiple cells in the other color.

## Surgical removal of *Drosophila* Malpighian tubule

Flies were anesthetized on a CO$_2$ pad prior to surgery, and were laid on their sides to expose the boundary between the dorsal and ventral abdomen. The A1 abdominal pleura was opened up with fine forceps (Fine Science Tools #5SF) under a stereoscopic microscope and one of the anterior pair of Malpighian tubules was pulled out and severed at appropriate sites as indicated in the figure legends. After surgery, flies were transferred to regular food for recovery.

## Flip-out lacZ marked lineage tracing

To conduct Flip-out lacZ marked lineage tracing of RSCs after surgical injury in *Figure 3—figure supplement 1F-G*, *Act5C > STOP > lacZ* tracer flies were crossed to *esg-Gal4,UAS-flp,UAS-myrRFP; tub-Gal80^{ts}* driver flies at 18°C. 3–7 day old progeny flies with appropriate genotypes were subject to surgical removal of one of the Malpighian tubules at desired sites, as specified in the figure legends prior to being shifted to 29°C to initiate lineage labeling. In order to conduct Flip-out lacZ marked lineage tracing of RSCs expressing UAS-N^{RNAi} (*Figure 4D*), UAS-NICD or UAS-ct (*Figure 4—figure supplement 1B and C*), *Act5C > STOP > lacZ* was combined with the transgene of interest first. The resultant *UAS-X; Act5C > STOP > lacZ* tracer flies were then crossed to the *esg-Gal4,UAS-flp,UAS-myrRFP; tub-Gal80^{ts}* driver flies.

## Genetic cell ablation and recovery

Cell-type specific genetic ablation was achieved by crossing *UAS-rpr,UAS-hid;tub-Gal80^{ts}* with cell-type specific Gal4 lines. For ablation of principal cells in the SCZ, *c507-Gal4,UAS-myr::tdTomato* males were crossed with *UAS-rpr,UAS-hid;tub-Gal80^{ts}* females at 18°C. 3–5 day old *UAS-rpr,UAS-hid/+; tub-Gal80^{ts}/+; c507-Gal4,UAS-myr::tdTomato/+* females were then shifted to 29°C for 7 days. After that, the flies were shifted back to 18°C to minimize the expression of *rpr* and *hid* to recover. Similarly, *tsh-Gal4* and *Uro-Gal4* lines were used to ablate stellate cells and principal cells at the main segment of Malpighian tubules, respectively.

## Single cell RNA-seq

*Drosophila* Malpighian tubules from wild type (5–7 day-old Oregon-R) female flies and *c507-Gal4^{ts} > rpr,hid* females that underwent PC-ablation (7 days at 29°C) followed by recovery (21 days at 18°C) were dissected in nuclease-free PBS on ice. *Drosophila* gut and upper tubules were carefully removed. Ureter and lower tubules from ~200 females were collected within 2 hr and then transferred to an Eppendorf tube containing 500 µl of dissociation buffer (1 mg/ml collagenase, 0.5 mg/ml elastase in nuclease free PBS). After incubating at 27°C for 1 hr on a nutator, the digestion reaction was stopped using 100 µl of fetal bovine serum. The dissociated cell suspension was then passed through a 70 µm cell strainer. The cells were spun down at 500 g, 4°C for 10 min and the supernatant was removed. 1 ml ice cold PBS was added to briefly resuspend and rinse cell pellets. After rinsing, the cells were spun down and resuspended with 100 µl of nuclease-free PBS.

Cells from wild type SCZ were encapsulated and the cDNA library was prepared at Genetic Resources Core Facility at the Johns Hopkins School of Medicine using Chromium Single Cell 3' Reagent Kits (v2). Cells from regenerated SCZ were encapsulated and the cDNA library was prepared at the Carnegie Institution using Chromium Single Cell 3' Reagent Kits (v3.1). The cDNA libraries were sequenced on an Illumina NextSeq 500 sequencer at the Carnegie Institution. Reads were mapped to the *Drosophila melanogaster* genome BDGP6.22 (Ensemble release 98) using CellRanger (10x Genomics Inc version 3.1.0). R package Seurat (v3.1.1) was used to analyze single cell RNA-seq results (*Stuart et al., 2019*). For wild type SCZ scRNA-seq, 710 cells were retained with the following metrics: 1) percentage of mitochondria genes < 30%; 2) 200 < nFeature_RNA < 4000; 3) 5000 < nCount_RNA < 10000. For regenerated SCZ scRNA-seq, 1499 cells were retained with the following metrics: 1) percentage of mitochondria genes < 30%; 2) 200 < nFeature_RNA < 5000; 3) 4500 < nCount_RNA < 15000. To identify common cell types between wild type and regenerated tissues, the two datasets were integrated together and integ rated analysis were performed on all cells using Seurat v3.1.1 (*Stuart et al., 2019*).

## EdU incorporation

Depending on the design of experiments, different EdU pulse-chase strategies were used to assay EdU incorporation. To assay EdU incorporation in flies under normal condition, 3–5 day old females (*esg-Gal4,UAS-myr::tdTomato*) were fed on standard cornmeal molasses food supplemented with 0.5 mM EdU (Click-iT EdU Alexa Fluor 488 Imaging Kit, ThermoFisher) for 2–4 days. EdU incorporation were detected on day 2 and day 4 following manufacturer's instructions. To assay EdU incorporation in flies after surgical removal of upper Malpighian tubules, 3–5 day old females (*esg-Gal4, UAS-myr::tdTomato*) were first fed on EdU food for 2 days and were continuously cultured on EdU food for two more days after amputation of Malpighian tubules.

## Bulk mRNA-seq

*Drosophila* ureter and lower tubules were dissected from ~100 females in nuclease-free PBS on ice. Gut and upper tubules were carefully removed. The isolated tissues were transferred to 500 µl of ice cold TRIzol (Invitrogen 15596026). the RNAs were extracted following manufacturer's instructions. cDNA libraries were prepared from poly(A)-selected RNA using Illumina TruSeq RNA Library Prep Kit v2, and sequenced using an Illumina Nextseq 500 sequencer. Single-end reads were mapped to the *Drosophila melanogaster* genome (dm6) using HISAT2 2.1.0 , transcripts were assembled using StringTie. Reads aligned to genes were calculated using featureCount (*Liao et al., 2014*) and differentially expressed genes were identified using R package DESeq2 (*Love et al., 2014*).

## X-Gal staining

Malpighian tubules were dissected into ice cold PBS and fixed in 0.5% Glutaraldehyde/PBS at room temperature for 15 min. After briefly wash in PBS, the samples were transferred into 1 ml of staining solution (10 mM $NaH_2PO_4 \cdot H_2O$, 10 mM $Na_2HPO_4 \cdot 2H_2O$, 150 mM NaCl, 1 mM $MgCl_2 \cdot 6H_2O$, 3.1 mM $K_4[Fe(CN)_6] \cdot 3H2O$, 3.1 mM $K_3[Fe(CN)_6]$, 0.3% TritonX-100, 0.2% X-Gal) and incubated overnight at room temperature.

## Nuclear volume measurement

z-stack images were acquired using a Leica TCS SP8 confocal microscope with a 20X objective lens (NA = 0.75). 3D images were reconstructed using Bitplane IMARIS v9.2.1 (http://www.bitplane.com/). Surface reconstruction of the DNA (DAPI staining) was created using surface tool of IMARIS. Nuclear volume was calculated by IMARIS in the surface statistics. Given that nuclear DNA is proportional to nuclear volume, cell ploidy was estimated by measuring nuclear volume based on DAPI staining. The nuclear volume of the cycling intestinal stem cells which have 2 C-4C DNA content was measured as a reference.

## Cell number quantification

For the quantification of esg+ cell number, z-stack images were acquired using a Leica TCS SP8 confocal microscope with a 20X objective lens (NA = 0.75) and were reconstructed into 3D image using IMARIS v9.2.1. The esg >RFP channel was used for automated IMARIS spots detection (set the estimated XY diameter = 5 µm). The results were verified manually and additional spots were added or removed if necessary. For the quantification of total cells in the SCZ as in *Figure 7—figure supplement 1D*, maximum intensity projection images of z-stacked DAPI channel were generated and automated cell counting was performed using ImageJ particle analyzer.

## Severity classification of stones in *Drosophila* Malpighian tubules

In *Figure 7—figure supplement 1*, stones were arbitrarily categorized into three classes based on the severity. Class I indicates stones are present in the SCZ in only one of the four MTs. Class II indicates stones are present in the SCZ in two of the four MTs. Class III indicates stones are present in the SCZ in at least three of the four MTs.

## Lifespan measurement

Wandering L3 larvae of control (*esg^{ts} >RFP*) and *esg^{ts} >RFP + Rac1.N17* were sorted and shifted from 18°C to 29°C. After eclosion, groups of 15–20 females of each genotype were placed into separate vials with normal cornmeal food or Allopurinol-augmented food and maintained at 29°C. The flies were transferred to new food every other day. The number of dead flies was recorded and the number of flies that escaped during transfer was excluded from the study. The statistical significance was determined by log-rank tests. ns denotes $p>0.05$, * denotes $p<0.05$, ** denotes $p<0.01$, *** denotes $p<0.001$.

## Acknowledgements

We are grateful to Jennifer Urban and Xin Chen (Johns Hopkins University), the Bloomington *Drosophila* Stock Center, Vienna *Drosophila* Resource Center and Developmental Studies Hybridoma Bank for *Drosophila* strains and reagents. We thank Allison Pinder and Frederick Tan for assistance

in RNA sequencing and analysis. We thank Joshua Blundon and Taylar Mouton for assistance in collecting samples for single cell RNA-seq. We are grateful to Mike Sepanski for help in electron microscopy. We thank Steve DeLuca, Ethan Greenblatt, Robert Levis, and other members of the Spradling laboratory for providing comments on the manuscript.

## Additional information

### Funding

| Funder | Grant reference number | Author |
|---|---|---|
| Howard Hughes Medical Institute | Allan Spradling | Allan C Spradling |

The funders had no role in study design, data collection and interpretation, or the decision to submit the work for publication.

### Author contributions

Chenhui Wang, Conceptualization, Data curation, Formal analysis, Validation, Investigation, Visualization, Methodology, Writing - original draft, Writing - review and editing; Allan C Spradling, Conceptualization, Supervision, Funding acquisition, Writing - original draft, Writing - review and editing

### Author ORCIDs

Chenhui Wang (iD) https://orcid.org/0000-0002-7408-234X
Allan C Spradling (iD) https://orcid.org/0000-0002-5251-1801

### Decision letter and Author response

Decision letter https://doi.org/10.7554/eLife.54096.sa1
Author response https://doi.org/10.7554/eLife.54096.sa2

## Additional files

### Supplementary files

- Supplementary file 1. Key resources table.

- Transparent reporting form

### Data availability

Data submitted to NCBI under accession code PRJNA595625.

The following dataset was generated:

| Author(s) | Year | Dataset title | Dataset URL | Database and Identifier |
|---|---|---|---|---|
| Wang C, Spradling A | 2020 | Transcriptional profiling of Drosophila ureter and lower tubules | http://www.ncbi.nlm.nih.gov/bioproject/PRJNA595625 | NCBI BioProject, PRJNA595625 |

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
