## [Decision Letter]

**Acceptance summary:**

In this manuscript, Wang and Spradling characterize a quiescent (cell-cycle arrested) renal stem cell population in *Drosophila* Malpighian tubules. These stem cells are unipotent, and their proliferation upon surgical damage or allopurinol induced stone formation serves to replace lost or damaged principal cells. This study helps to establish the Malpighian tubules as an important model for studying injury induced stem cell activation.

**Decision letter after peer review:**

Thank you for submitting your article "An abundant quiescent stem cell population in *Drosophila* Malpighian tubules protects principal cells from kidney stones" for consideration by *eLife*. Your article has been reviewed by four peer reviewers, including Michael Buszczak as the Reviewing Editor and Reviewer #1, and the evaluation has been overseen by Anna Akhmanova as the Senior Editor.

The reviewers have discussed the reviews with one another and the Reviewing Editor has drafted this decision to help you prepare a revised submission.

Summary:

In this manuscript, Wang and Spradling describe their characterization of a stem cell population in *Drosophila* Malpighian tubules. They start by determining the relative number of renal stem cells (RSCs) compared to two populations of principal cells (PCs), which can be distinguished based on differences in their ploidy. The authors then perform single cell RNA-seq analysis and identify 6 distinct populations of cells based on gene expression patterns. These include the RSCs, three distinct population of PCs, Stellate cells and a small sixth population that remains largely uncharacterized. Genetic ablation and lineage experiments show that the RSCs replenish PCs in the SCZ but do not give rise to stellate cells or PCs in the upper tubules. The authors develop a surgical method to specifically damage different regions of the Malpighian tubules. They find without damage, the RSCs are largely inactive but start to divide upon local damage. Damage induces upregulation of a number of short range signaling pathways, including the Jak/Stat, EGF and Notch pathways. The authors use lineage tracing to show that RSCs can exhibit symmetric or asymmetric divisions upon damage and that they are unipotent, which runs counter to previously published data. Finally, the authors show RSCs respond to damage caused by kidney stones.

This is an interesting and data-rich study that serves to further establish *Drosophila* Malpighian tubules as a useful stem cell model system. In general, the data are convincing and nicely presented. Importantly, this study provides robust data that RSCs are in fact unipotent and greatly increase their activity upon different forms of damage. The authors also succeed in further developing a number of damage paradigms that will be of use to the field. The study is potentially appropriate for publication in *eLife* and will be of broad interest to the scientific community. However, the points listed below need to be addressed before acceptance of the manuscript.

Essential revisions:

1) The authors conclude that RSCs are largely quiescent, but low levels of cell division is only one aspect of this cell state. Quiescent cells also exhibit very low levels of transcription and translation, and low mitochondrial activity. The authors need to provide further evidence that RSCs in undamaged Malpighian tubules have these characteristics of quiescence, or else tone down their claims that RSCs are quiescent in both the manuscript title and throughout the text. For example, in Figure 1, the authors provide EM data that show RSCs have fewer mitochondria with fewer cristae relative to differentiated PCs. These data are used to establish the idea that RSCs normally reside in a quiescent state. However, this level of analysis is not satisfying. The authors should characterize mitochondrial activity, and complement their EM data with immunofluorescence to further verify that mitochondrial numbers differ between RSCs and PCs. Importantly, all of these data should be robustly quantified. Further experiments to show clear differences in transcription and/or translation between RSCs and differentiated cells would also strengthen these conclusions.

2) A major claim of the paper is that new principal cells form as a result of renal stem cell division giving rise to renoblasts followed by differentiation into principal cells. However, more could be done to convince the reader that these are, indeed, differentiated principal cells; the argument rests on higher levels of Cut positivity and the increased ploidy of the putative principal cells as compared to the RSCs/renoblasts. However, the ploidy remains less than seen in the pre-existing principal cells, and whether the new principal cells have other features of mature/differentiated principal cells is not explored. In addition, the single cell RNAseq data show that there is another Cut+ cell type that also express genes characteristic of stellate cells (the transcription factor tsh and the aquaporin Drip), raising the question of whether the cut+ replacement cells are RSCs, differentiated PCs, or perhaps even stellate (or stellate-like) cells. (The "replacement principal cells" have small nuclei, not unlike differentiated stellate cells found in the main segment.) Further characterization of these Cut+ cells, and particularly the expression of markers of differentiated principal cells that are not expressed in renal stem cells, would be helpful.

3) The authors conclude the Results section by stating that "RSCs are required to generate new replacement PCs in response to damage and that RSC-mediated regeneration is advantageous for survival under the conditions of kidney stress." This is not definitively proven. In addition to the concerns raised above, in the stone model the phenotype that is demonstrated is a proliferative response, but not "regeneration." Second, the absence of RSCs could have deleterious consequences for reasons other than an absence of regeneration; interfering with regeneration in other ways (for example, by manipulating one or more of the pathways required for the RSC to renoblast progression) would be more convincing. Furthermore, since stem cell zone RSC depletion is achieved by expressing dominant-negative Rac using *esg-GAL4*, one wonders whether there are effects on the midgut that are contributing to lethality. ie, impaired ISCs/messed up midguts + stone stress in tubules could lead to lethality independent of the failure of the tubule to respond to the stone stress. In addition, as an explanation of why RSCs would be restricted to the ureter and lower tubules, the authors offer the explanation that xanthine stones often start to form in the ureter and lower tubules. But normally these stones are prevented by the presence of rosy (encoding xanthine dehydrogenase), and flies presumably are not frequently coming into contact with allopurinol, so it's unclear how much evolutionary pressure this would exert. Indeed, why the proliferative/regenerative potential of the tubule is limited to the lower tubule is mysterious, given the importance of the upstream segments for renal function. Can the animal live very long without stellate cells and/or PCs? The authors should address these points experimentally and provide better explanations in the text.

4) A number of previous studies have described how RSCs are regulated by both the Notch and EGFR pathways (Li et al., 2014, Further, differential Notch activity is required for homeostasis of Malpighian tubules in adult *Drosophila* J Genet Genomics 41:649-652; Li et al., 2015, EGFR/MAPK signaling regulates the proliferation of *Drosophila* renal and nephric stem cells. J Genet Genomics 42:9e20; reviewed in Gautam et al., 2017). Although the Gautam review is cited, the two previous Li papers are not. The authors should cite these papers and fully describe the data therein in the Introduction and/or Results section so that their own data can be placed within this context.

[Editors' note: further revisions were suggested prior to acceptance, as described below.]

Thank you for resubmitting your work entitled "An abundant quiescent stem cell population in *Drosophila* Malpighian tubules protects principal cells from kidney stones" for further consideration by *eLife*. Your revised article has been evaluated by Anna Akhmanova (Senior Editor) and a Reviewing Editor.

The manuscript has been improved but there are three remaining issues that need to be addressed before acceptance, as outlined below:

1) The authors uploaded new figures into the system but kept the old figures embedded in the text file. To avoid any confusion, they should delete the old images. Note – reviewers 3's comments below were directed at the old figures within the text.

2) No information is provided as to whether the combination of *esg>Rac1.N17* with allopurinol affects the midgut in ways that could contribute to the shorter lifespan seen in the allopurinol-treated *esg>Rac1.N17* flies. This deficiency could be addressed in one of two ways. First, the midguts of allopurinol-treated *esg>Rac1.N17* flies could be studied. Although morphology is not equivalent to function, demonstrating normal midgut morphology in these flies would strengthen the argument that the deleterious effects are limited to the tubule. Alternatively, the penultimate sentence of subsection “Modeling kidney stone disease in *Drosophila*” could be modified to read along the lines of: "Additionally, preventing RSC-mediated regeneration of the SCZ extends lifespan, although we cannot exclude effects on the midgut of expressing Rac1.N17 in esg+ ISCs/RSCs.”

3) Figure 1—figure supplement 1 is missing a panel between G and H showing what Allopurinol alone looks like.

The individual comments from each of the reviewers are listed below for your consideration.

Reviewer #1:

In this revised manuscript, the authors have successfully addressed most of the comments from the first round of reviews. The addition of new sequencing data, further quantification of existing data, citations to previously published work, and clarification of various points throughout the text have greatly improved the manuscript.

However, the authors have not addressed the previous concern that *esg>Rac1.N17* + allopurinol results in damage to the midgut, which could complicate the interpretation of the lifespan data presented in Figure 7J. This is somewhat of a minor point. The authors could either examine whether they observe damage in the midguts of *esg>Rac1.N17* + allopurinol treated flies or slightly soften their concluding sentence (final paragraph of the Results section). Knock-down of cut, which prevents the formation of new PCs in the presence of allopurinol (Figure 4—figure supplement 1H), may represent another way to assess whether PC regeneration in the SCZ extends the longevity of stone-carrying flies.

In addition, the authors kept the old versions of the figures embedded in the corrected text. To avoid any confusion moving forward, they authors should eliminate these old figures.

Reviewer #2:

New data solidify arguments that replacement principal cells are transcriptionally similar to the original principal cells, and thus that RSCs are indeed regenerating principal cells; and that the RSCs do not give rise to progeny populating the initial/transitional/main segments, both of which are important findings. They have tested necessity (in addition to previous experiments testing sufficiency) of cut in allopurinol-treated flies which helps support the model in Figure 4I, but are missing a control of allopurinol alone without cut RNAi.

Points to address:

1) Authors have now added in quantification of several findings, and perform statistics for some but not all of the quantified data (for example Figure 1F, G, Figure 3 D, I, L, Figure 3—figure supplement 2E).

2) The authors state in the revised manuscript that it is "commonly believed that the main segment of the Malpighian tubules is responsible for fluid secretion, whereas the ureter and lower tubules are reabsorptive." This "belief" There are data to support this assertion, eg Dow et al, 1994 and O'Donnell and Maddrell, 1995.

3) Figure 2 shows a small Drip^+^ cluster of cells. In the response to reviewers, the authors argue that this may represent some contamination of stellate cell RNA from the main segment, doublets etc. and therefore they have labeled these as "stellate?" cells. Would be worth having a sentence in the results stating the authors' conclusion that there are unlikely to be resident stellate cells in the lower segment, as shown by others and confirmed by the authors in Figure 1—figure supplement 1F, and that the apparent presence of a small population of Drip^+^ cells likely represents contamination. (This is mentioned later in Figure 3 supplement, but would be worth mentioning earlier.)

4) The new supplemental data in Figure 3—figure supplement 1F are convincing in making the point that the replacement PCs are transcriptionally similar to the original PCs; suggest moving into the main figures if there's space.

5) Specify in figure legends what is being stained in Figure 3 A-C.

6) Figure 4—figure supplement 1 is missing a panel between G and H showing what Allopurinol alone looks like.

7) Figure 7—figure supplement 1: image of class I, II and III stones are shown. The Materials and methods should specify how these classes are defined.

8) The authors acknowledge in their rebuttal that "the incidence of kidney stones in wild populations is not known." Therefore, it is a stretch to say that "the deployment of RSCs in the SCZ is driven by evolutionary selection pressure for regenerative capacity in regions where stones are most likely". Also, while data from this paper suggest that xanthine stones seem to have a propensity for forming in proximity to the SCZ, this is not true for all stone types in the Malpighian tubule; see for example PMID 22993075, PMID 21451462, and PMID 22352299. In fact, work from others shows that xanthine dehydrogenase knockdown results in stones extensively throughout the tubules – see Figures 1A and 3A in PMID 25970330. It is possible, however, that the steric constraints found where the two tubules converge at the ureter do make this region more likely to suffer epithelial damage. Could these factors also make this part of the tubule particularly susceptible to heat stress? Suggest a slight rewording of this sentence in the Discussion, along the lines of: "the deployment of RSCs in the SCZ could be driven by evolutionary selection pressure for regenerative capacity in regions where epithelial damage from stones or other stressors could occur."

9) I am still left wondering about whether Rac1.N17 expression in esg+ progenitors has any effect on the midgut. No information is provided as to whether the combination of *esg>Rac1.N17* with allopurinol affects the midgut in ways that could contribute to the shorter lifespan seen in the allopurinol-treated *esg>Rac1.N17* flies. This deficiency could be addressed in one of two ways. First, the midguts of allopurinol-treated *esg>Rac1.N17* flies could be studied. Although morphology is not equivalent to function, demonstrating normal midgut morphology in these flies would strengthen the argument that the deleterious effects are limited to the tubule. Alternatively, the penultimate sentence of subsection “Modeling kidney stone disease in *Drosophila*” could be modified to read along the lines of: "Additionally, preventing RSC-mediated regeneration of the SCZ extends lifespan, although we cannot exclude effects on the midgut of expressing Rac1.N17 in esg+ ISCs/RSCs."

Reviewer #3:

There are still many issues, which make it difficult for me to assess whether all of my concerns were addressed. The authors are encouraged to please double check all text and figures to ensure correctness.

In the legend to Figure 1, all of the panels are mis-labelled.

In Figure 3, the legend talks about panels K and L but there are no K and L panels in the figure. Also Figure 3I is not a quantification.

In Figure 5, the legend for C is wrong and then the legend for D-F is wrong. There is no panel G.

In Figure 6, there is no dashed red box.

In Figure 7, there are no outlined boxes in E and G.

In the main body text, many figures are incorrectly cited.

Figure 1: I can't see the labels for C1 and C2 because of black font on a gray background.

---

## [Author Response]

Essential revisions:1) The authors conclude that RSCs are largely quiescent, but low levels of cell division is only one aspect of this cell state. Quiescent cells also exhibit very low levels of transcription and translation, and low mitochondrial activity. The authors need to provide further evidence that RSCs in undamaged Malpighian tubules have these characteristics of quiescence, or else tone down their claims that RSCs are quiescent in both the manuscript title and throughout the text. For example, in Figure 1, the authors provide EM data that show RSCs have fewer mitochondria with fewer cristae relative to differentiated PCs. These data are used to establish the idea that RSCs normally reside in a quiescent state. However, this level of analysis is not satisfying. The authors should characterize mitochondrial activity, and complement their EM data with immunofluorescence to further verify that mitochondrial numbers differ between RSCs and PCs. Importantly, all of these data should be robustly quantified. Further experiments to show clear differences in transcription and/or translation between RSCs and differentiated cells would also strengthen these conclusions.

We never meant to make the metabolic and/or gene activity state of the RSCs a major focus of the paper. So we have elected to “tone down” what appeared to the reviewers to be unsupported claims in this regard, something that was not our intention. By "quiescent," we simply meant that RSCs are cell cycle arrested, unlike intestinal stem cells (ISCs) or germline stem cells (GSCs) which actively produce progeny to replace cell loss. We followed the conventional terminology in the field and thus named the population of stem cells as “quiescent stem cells”. The difference in mitochondrial number seen in EMs is a general "throw away" type observation seen with many types of stem cells, and is a very general characteristic of less differentiated cells, from which we draw no strong conclusions. We do not have a position on the level of RSC metabolic activity, since even non-dividing "quiescent" cells such as stage14 oocytes are a lot more active in protein synthesis than used to be thought (Greenblatt and Spradling, 2019). Consequently, to avoid misunderstanding we removed the statement that RSCs have fewer mitochondria than PCs and we have now defined the word “quiescent” as meaning cell cycle arrested in the Abstract and Introduction section. The reviewers pointed out new research directions which are potentially exciting and worth further investigation in the future.

2) A major claim of the paper is that new principal cells form as a result of renal stem cell division giving rise to renoblasts followed by differentiation into principal cells. However, more could be done to convince the reader that these are, indeed, differentiated principal cells; the argument rests on higher levels of Cut positivity and the increased ploidy of the putative principal cells as compared to the RSCs/renoblasts. However, the ploidy remains less than seen in the pre-existing principal cells, and whether the new principal cells have other features of mature/differentiated principal cells is not explored. In addition, the single cell RNAseq data show that there is another Cut+ cell type that also express genes characteristic of stellate cells (the transcription factor tsh and the aquaporin Drip), raising the question of whether the cut+ replacement cells are RSCs, differentiated PCs, or perhaps even stellate (or stellate-like) cells. (The "replacement principal cells" have small nuclei, not unlike differentiated stellate cells found in the main segment.) Further characterization of these Cut+ cells, and particularly the expression of markers of differentiated principal cells that are not expressed in renal stem cells, would be helpful.

We have now provided single cell RNA-seq data of the regenerated SCZ from *c507-Gal4^ts^ >UAS-rpr,hid* females (21-day recovery at 18^o^C after expression of *rpr,hid* at 29^o^C for 7 days). To eliminate batch effect of scRNA-seq, we did integrated analysis of scRNA-seq data across wild type and regenerated tissues (New Figure 3—figure supplement 1F). Same cell types were clustered together after integrated analysis. We can identify principal cells in both wild type and regenerated SCZ by integrated analysis. The replacement principal cells are very similar to the preexisting principal cells in their transcriptome. We verified that by examining the expression of a list of genes that are highly enriched in differentiated principal cells, indicating the replacement cells are true PCs in spite of their distinct nuclear size.

In addition, we ruled out the possibility that RSCs gave rise to stellate cells in the SCZ after PC-ablation by examining the expression of *tsh-lacZ*. The stellate cells marked by *tsh-lacZ* are only present in the upper tubules in both wild type and regenerated Malpighian tubules (New Figure 3—figure supplement 1C-D), indicating the replacement cells are not stellate cells. And the stellate cell cluster in our scRNAseq data contains few cells, and its presence at all is most likely due to contamination of our starting population of dissected lower tubules with a few Malpighian tubule upper segment regions that contain true stellate cells.

Finally, we found that Cut protein could not be detected in stellate cells (New Figure 3—figure supplement 1E), suggesting the Cut mRNA detected in the “stellate cells” might have different sources in the scRNA-seq. It could be due to ambient RNA from dead principal cells (they are probably more breakable than other cells during preparation due to their very large size), or else that principal cells and stellate cells formed doublets. Indeed, after further filtering out the cells with a clear outlier number of genes detected, the expression of Cut mRNA is barely detectable in the Drip^+^ cell clusters (revised Figure 2C).

3) The authors conclude the Results section by stating that "RSCs are required to generate new replacement PCs in response to damage and that RSC-mediated regeneration is advantageous for survival under the conditions of kidney stress." This is not definitively proven. In addition to the concerns raised above, in the stone model the phenotype that is demonstrated is a proliferative response, but not "regeneration."

We believe that we have addressed the reviewers’ previous concerns that the small replacement cells might not be PCs, justifying our conclusion that RSCs generate replacement PCs in response to PC damage. We showed by clonal analysis that RSCs generate the regenerated PCs. We showed that new PCs do not arise in MT regions lacking RSCs or more than 10 PC diameters away from the closest RSC. We genetically ablated RSCs and saw that regeneration was blocked. This level of evidence justifies our statement about that RSCs are required to generate new PCs.

We do not agree that in the stone model all we have shown is a proliferative response. We showed that stones damage PCs and induce PC regeneration, not just proliferation. This includes evidence that stones induced activation of both proliferation pathways (JNK, Jak/STAT, EGFR/MAPK, Hippo/Yki) and differentiation pathways (Notch pathway) (Figure 7—figure supplement 2), just like other types of damage (PC-ablation, surgical injury). Lineage tracing with MARCM system showed RSCs gave rise to Cut^+^ replacement principal cells with smaller ploidy in Allopurinol treated animals (Figure 4E and 4G). Consistently, *esg-Gal4^ts^* directed F/O-lacZ lineage tracing studies also showed formation of new tubule cells (*esg-lacZ^+^*, denoted by arrows) in the Malpighian tubules of Allopurinol treated animals (Author response image 1). Because stone-activated RSCs can proliferate and differentiate, it is not only a proliferative response but also PC regeneration.

**Author response image 1. respfig1:** Kidney stones lead to regeneration of the SCZ.

Second, the absence of RSCs could have deleterious consequences for reasons other than an absence of regeneration; interfering with regeneration in other ways (for example, by manipulating one or more of the pathways required for the RSC to renoblast progression) would be more convincing. Furthermore, since stem cell zone RSC depletion is achieved by expressing dominant-negative Rac using esg-GAL4, one wonders whether there are effects on the midgut that are contributing to lethality. ie, impaired ISCs/messed up midguts + stone stress in tubules could lead to lethality independent of the failure of the tubule to respond to the stone stress.

The reviewers’ argument that *esg>Rac1.N17* expression could be directly damaging ISCs or midguts and affecting longevity is not a very strong one. There was no difference in longevity due to expressing *esg>Rac1.N17* in the absence of Allopurinol (Figure 7J). The fact that there is a significant difference in survival without RSCs following stone induction using allopurinol treatment, but no effect of RSC loss without allopurinol treatment and stones, we believe justifies our statement (Figure 7J). The reviewers suggest that we further test the role of RSC-mediated regeneration by interfering with RB progression rather than by eliminating RSCs themselves. However, there is no Gal4-driver that is specifically expressed in RSCs or RBs but not in ISCs or EBs. Therefore, we cannot disrupt RB progression without disrupting EB progress at the same time, which would not be interpretable.

In addition, as an explanation of why RSCs would be restricted to the ureter and lower tubules, the authors offer the explanation that xanthine stones often start to form in the ureter and lower tubules. But normally these stones are prevented by the presence of rosy (encoding xanthine dehydrogenase), and flies presumably are not frequently coming into contact with allopurinol, so it's unclear how much evolutionary pressure this would exert. Indeed, why the proliferative/regenerative potential of the tubule is limited to the lower tubule is mysterious, given the importance of the upstream segments for renal function. Can the animal live very long without stellate cells and/or PCs? The authors should address these points experimentally and provide better explanations in the text.

Regarding why the tubules only regenerate PCs but not stellate cells, and why RSCs are confined to the lower segments- these are evolutionary issues that cannot be addressed experimentally and are destined to remain a subject of speculation. The suggested experiments would not be informative as many essential cell types in *Drosophila* (like in humans) do not undergo regeneration. We think it highly likely that lack of PCs and Stellate cellswould shortenthe life span of *Drosophila*. The incidence of kidney stones in wild populations is not known since the lab conditions do not represent the natural living conditions for wild populations (Markow, 2015, *eLife*). That is to say, the evolutionary selective pressure could be very different between lab and wild flies.

Thus, the reviewers cannot assume that they are inconsequential just because we used genetic or pharmacological means to study them. In contrast, the fact that flies contain an abundant, quiescent stem cell population that can regenerate PCs is a strong argument that these cells are either particularly sensitive to damage or particularly important to maintain. We also uncovered a potential alternative evolutionary force, namely that heat stress damages principal cells, based on evidence in the literature and in our experiments for such an effect. Heat stress is very likely a factor in wild populations.

4) A number of previous studies have described how RSCs are regulated by both the Notch and EGFR pathways (Li et al., 2014, Further, differential Notch activity is required for homeostasis of Malpighian tubules in adult *Drosophila* J Genet Genomics 41:649-652; Li et al., 2015, EGFR/MAPK signaling regulates the proliferation of *Drosophila* renal and nephric stem cells. J Genet Genomics 42:9e20; reviewed in Gautam et al., 2017,). Although the Gautam review is cited, the two previous Li papers are not. The authors should cite these papers and fully describe the data therein in the Introduction and/or Results section so that their own data can be placed within this context.

We have added and discussed these references, which we do not believe significantly anticipate the results presented here.

[Editors' note: further revisions were suggested prior to acceptance, as described below.]

The manuscript has been improved but there are three remaining issues that need to be addressed before acceptance, as outlined below:1) The authors uploaded new figures into the system but kept the old figures embedded in the text file. To avoid any confusion, they should delete the old images. Note reviewers 3's comments below were directed at the old figures within the text.

We have deleted all the old figures to avoid confusion.

2) No information is provided as to whether the combination of esg>Rac1.N17 with allopurinol affects the midgut in ways that could contribute to the shorter lifespan seen in the allopurinol-treated esg>Rac1.N17 flies. This deficiency could be addressed in one of two ways. First, the midguts of allopurinol-treated esg>Rac1.N17 flies could be studied. Although morphology is not equivalent to function, demonstrating normal midgut morphology in these flies would strengthen the argument that the deleterious effects are limited to the tubule. Alternatively, the penultimate sentence of subsection “Modeling kidney stone disease in *Drosophila*” could be modified to read along the lines of: "Additionally, preventing RSC-mediated regeneration of the SCZ extends lifespan, although we cannot exclude effects on the midgut of expressing Rac1.N17 in esg+ ISCs/RSCs.”

We agree that we did not completely rule out that effects on the midgut contribute along with the loss of RSC-mediated repair to the shortened life span in the presence of allopurinol, but it seems unlikely such effects could be identified simply by histology. We therefore added a qualification along the lines recommended to our statement: “Additionally, preventing RSC-mediated regeneration of the SCZ shortens lifespan in the presence of allopurinol-induced kidney stones, although we cannot exclude effects on the midgut of expressing Rac1.N17 in esg+ ISCs/RSCs.”

3) Figure 1—figure supplement 1 is missing a panel between G and H showing what Allopurinol alone looks like.

We have now provided the Allopurinol alone panel in the new Figure 4—figure supplement 1.

The individual comments from each of the reviewers are listed below for your consideration.Reviewer #1:In this revised manuscript, the authors have successfully addressed most of the comments from the first round of reviews. The addition of new sequencing data, further quantification of existing data, citations to previously published work, and clarification of various points throughout the text have greatly improved the manuscript.However, the authors have not addressed the previous concern that esg>Rac1.N17 + allopurinol results in damage to the midgut, which could complicate the interpretation of the lifespan data presented in Figure 7J. This is somewhat of a minor point. The authors could either examine whether they observe damage in the midguts of esg>Rac1.N17 + allopurinol treated flies or slightly soften their concluding sentence (final paragraph of the Results section). Knock-down of cut, which prevents the formation of new PCs in the presence of allopurinol (Figure 4—figure supplement 1H), may represent another way to assess whether PC regeneration in the SCZ extends the longevity of stone-carrying flies.In addition, the authors kept the old versions of the figures embedded in the corrected text. To avoid any confusion moving forward, they authors should eliminate these old figures.

We have now softened our conclusion by the addition of the word “apparently” and qualified our statement in the Discussion section. All old figures have been deleted to avoid confusion in the revised manuscript now.

Reviewer #2:New data solidify arguments that replacement principal cells are transcriptionally similar to the original principal cells, and thus that RSCs are indeed regenerating principal cells; and that the RSCs do not give rise to progeny populating the initial/transitional/main segments, both of which are important findings. They have tested necessity (in addition to previous experiments testing sufficiency) of cut in allopurinol-treated flies which helps support the model in Figure 4I, but are missing a control of allopurinol alone without cut RNAi.

We have now provided the Allopurinol alone panel in the new Figure 4—figure supplement 1.

Points to address:1) Authors have now added in quantification of several findings, and perform statistics for some but not all of the quantified data (for example Figure 1F, G, Figure 3 D, I, L, Figure 3—figure supplement 2E).

Figure 1F reports the average nuclear volume (related to ploidy) for three cell populations, but statistical comparison was not done since membership in the groups is determined by rules. We added statistical comparison to Figure 1G. We added statistical comparison to Figure 3D. We added statistical comparison to old Figure 3I, but this is now Figure 3J, and to old Figure 3L, but this is now Figure 3—figure supplement 1H. We added statistical comparison to Figure 3—figure supplement 2E.

2) The authors state in the revised manuscript that it is "commonly believed that the main segment of the Malpighian tubules is responsible for fluid secretion, whereas the ureter and lower tubules are reabsorptive." This "belief" There are data to support this assertion, eg Dow et al., 1994 and O'Donnell and Maddrell, 1995.

We have now cited the suggested papers to support the common belief.

3) Figure 2 shows a small Drip^+^ cluster of cells. In the response to reviewers, the authors argue that this may represent some contamination of stellate cell RNA from the main segment, doublets etc. and therefore they have labeled these as "stellate?" cells. Would be worth having a sentence in the results stating the authors' conclusion that there are unlikely to be resident stellate cells in the lower segment, as shown by others and confirmed by the authors in Figure 1—figure supplement 1F, and that the apparent presence of a small population of Drip^+^ cells likely represents contamination. (This is mentioned later in Figure 3 supplement, but would be worth mentioning earlier.)

Agreed. We have now mentioned that the presence of a small population of Drip^+^ cells likely represents contamination in the Results section.

4) The new supplemental data in Figure 3—figure supplement 1F are convincing in making the point that the replacement PCs are transcriptionally similar to the original PCs; suggest moving into the main figures if there's space.

Agreed. We have now moved Figure 3—figure supplement 1F to the main Figure 3. To make room for that, we moved Figure 3 J-F to Figure 3—figure supplement 1.

5) Specify in figure legends what is being stained in Figure 3 A-C.

Thanks for pointing out the oversight. We have now added the labeling to the Figure panels.

6) Figure 4—figure supplement 1 is missing a panel between G and H showing what Allopurinol alone looks like.

We have now provided the Allopurinol alone panel in the new Figure 4—figure supplement 1.

7) Figure 7—figure supplement 1: image of class I, II and III stones are shown. The Materials and methods should specify how these classes are defined.

Thanks for pointing out the missing information. We have now provided the criterion used to define stone classes in the Materials and methods section.

8) The authors acknowledge in their rebuttal that "the incidence of kidney stones in wild populations is not known." Therefore, it is a stretch to say that "the deployment of RSCs in the SCZ is driven by evolutionary selection pressure for regenerative capacity in regions where stones are most likely". Also, while data from this paper suggest that xanthine stones seem to have a propensity for forming in proximity to the SCZ, this is not true for all stone types in the Malpighian tubule; see for example PMID 22993075, PMID 21451462, and PMID 22352299. In fact, work from others shows that xanthine dehydrogenase knockdown results in stones extensively throughout the tubules – see Figures 1A and 3A in PMID 25970330. It is possible, however, that the steric constraints found where the two tubules converge at the ureter do make this region more likely to suffer epithelial damage. Could these factors also make this part of the tubule particularly susceptible to heat stress? Suggest a slight rewording of this sentence in the Discussion, along the lines of: "the deployment of RSCs in the SCZ could be driven by evolutionary selection pressure for regenerative capacity in regions where epithelial damage from stones or other stressors could occur."

The sentence reads "… suggesting that the deployment of RSCs.….". We are making an hypothesis to explain the presence of RSCs only in the SCZ based on our experiments showing preferential localization of Xanthine stones in this region.

9) I am still left wondering about whether Rac1.N17 expression in esg+ progenitors has any effect on the midgut. No information is provided as to whether the combination of esg>Rac1.N17 with allopurinol affects the midgut in ways that could contribute to the shorter lifespan seen in the allopurinol-treated esg>Rac1.N17 flies. This deficiency could be addressed in one of two ways. First, the midguts of allopurinol-treated esg>Rac1.N17 flies could be studied. Although morphology is not equivalent to function, demonstrating normal midgut morphology in these flies would strengthen the argument that the deleterious effects are limited to the tubule. Alternatively, the penultimate sentence of subsection “Modeling kidney stone disease in *Drosophila*” could be modified to read along the lines of: "Additionally, preventing RSC-mediated regeneration of the SCZ extends lifespan, although we cannot exclude effects on the midgut of expressing Rac1.N17 in esg+ ISCs/RSCs."

We modified the sentence as described above.

Reviewer #3:There are still many issues, which make it difficult for me to assess whether all of my concerns were addressed. The authors are encouraged to please double check all text and figures to ensure correctness.In the legend to Figure 1, all of the panels are mis-labelled.In Figure 3, the legend talks about panels K and L but there are no K and L panels in the figure. Also Figure 3I is not a quantification.In Figure 5, the legend for C is wrong and then the legend for D-F is wrong. There is no panel G.In Figure 6, there is no dashed red box.In Figure 7, there are no outlined boxes in E and G.In the main body text, many figures are incorrectly cited.Figure 1: I can't see the labels for C1 and C2 because of black font on a gray background.

We kept track of all the changes that we had made during last revision, in order to make it easier for reviewers to compare the original versus revised figures. Unfortunately, it apparently had caused confusions. We have now removed all the old figures embedded in the text. We have double checked the figures and texts to make sure the figures are correctly cited.